# ODE-based Smoothing Neural Network for Reinforcement Learning Tasks

**Yinuo Wang**[1], **Wenxuan Wang**[1], **Xujie Song**[1], **Tong Liu**[1], **Yuming Yin**[2],
**Liangfa Chen**[3], **Likun Wang**[1], **Jingliang Duan**[1,3*], **Shengbo Eben Li**[1*]
[1] School of Vehicle and Mobility & College of AI, Tsinghua University
[2] School of Mechanical Engineering, Zhejiang University of Technology
[3] School of Mechanical Engineering, University of Science and Technology Beijing
`wyn23@mails.tsinghua.edu.cn, duanjl15@163.com, lishbo@tsinghua.edu.cn`

## Abstract

The smoothness of control actions is a significant challenge faced by deep reinforcement learning (RL) techniques in solving optimal control problems. Existing RL-trained policies tend to produce non-smooth actions due to high-frequency input noise and unconstrained Lipschitz constants in neural networks. This article presents a Smooth ODE (SmODE) network capable of simultaneously addressing both causes of unsmooth control actions, thereby enhancing policy performance and robustness under noise condition. We first design a smooth ODE neuron with first-order low-pass filtering expression, which can dynamically filter out high frequency noises of hidden state by a learnable state-based system time constant. Additionally, we construct a state-based mapping function, $g$, and theoretically demonstrate its capacity to control the ODE neuron's Lipschitz constant. Then, based on the above neuronal structure design, we further advanced the SmODE network serving as RL policy approximators. This network is compatible with most existing RL algorithms, offering improved adaptability compared to prior approaches. Various experiments show that our SmODE network demonstrates superior anti-interference capabilities and smoother action outputs than the multilayer perceptron and smooth network architectures like LipsNet.

## 1 Introduction

Recently, deep reinforcement learning (RL) has emerged as an effective method for solving optimal control problems in the physical world Guan et al. (2022); Peng et al. (2021); Kaufmann et al. (2023); Li (2023); Wang et al. (2024). RL algorithms commonly employ neural networks (NNs) to learn optimal control policies due to their universal approximation capabilities Sonoda & Murata (2017); Schäfer & Zimmermann (2006). However, in practical optimal control scenarios, the outputs of NNs are often sensitive to noise disturbances, as noted by Molchanov et al. (2019). Inadequately addressing this sensitivity can result in severe consequences. For instance, oscillations in control actions may cause drone crashes Shi et al. (2019), increased wear in robotic arm components Yu et al. (2021), and heightened safety risks in autonomous driving Wasala et al. (2020); Chen et al. (2021).

To optimize NN performance in optimal control scenarios, research has primarily concentrated on improving the smoothness of NN-based control systems. Current approaches can be classified into four principal categories: filtering methods, action penalty methods, adversarial perturbation methods, and network enhancement methods.

Filtering methods like Kalman and extended Kalman filtering Chen et al. (2023) effectively suppress noise and reduce output oscillation by estimating the current state from multi-step historical data. These methods work well with Gaussian noise but struggle with non-Gaussian noise. Particle filtering Wang et al. (2021), in contrast, samples directly from the probability density function to address nonlinear and non-Gaussian noise, making it more suitable for such environments. However, it is computationally intensive due to the need for many samples and can suffer from particle degeneracy, affecting its accuracy Daum & Huang (2011).

---

*Corresponding authors.

Action penalty methods penalize significant shifts in actions to enhance stability and smoothness during policy learning. Mysore et al. (2021) incorporated two regularization components within the policy loss function: one mitigates variance between consecutive actions over time, and another promotes action consistency across similar states. Similarly, Kobayashi (2022) introduced the L2C2 algorithm with dual losses: one for action congruence and another for coherence in the value function across similar states, adjusting action penalties based on value function congruity. While these methods improve stability and smoothness, fine-tuning hyperparameters without diminishing system performance is challenging.

Adversarial perturbation techniques aim to reduce oscillatory output actions by integrating optimized perturbation data during training. The main goal is to enhance the agent's resistance to noisy data Zhao et al. (2022), improving control effectiveness in unpredictable or noisy environments. Shen et al. (2020) employed projected gradient ascent to identify the most effective perturbation noise, maximizing action divergence under genuine and adversarial conditions. This approach effectively mitigates the oscillation issues caused by noise. However, the algorithm increases complexity by generating adversarial states, and it faces compatibility challenges with mainstream RL algorithms and limited generalizability.

The aforementioned methods each have their drawbacks: the filtering method necessitates multi-step historical data, action penalties may compromise control optimality, and adversarial perturbations complicate RL methods. Network enhancements add noise resistance directly to the NN through structural improvements, avoiding major modifications to the RL algorithm. Miyato et al. (2018) employed spectral normalization to reduce the NN's Lipschitz constant, enhancing smoothness. Similarly, Song et al. (2023) introduced LipsNet, which adaptively modulates the local Lipschitz constant, effectively dampening action oscillation Gouk et al. (2021). Nonetheless, with high observation noise, controlling the NN's Lipschitz constant alone inadequately suppresses action fluctuations. Additionally, the neural ordinary differential equation (ODE) network Chen et al. (2018); Hasani et al. (2021); Asikis et al. (2022); Hasani et al. (2022); Ruiz-Balet & Zuazua (2023), defined by ODEs, emerges as a promising approach due to its flexibility in autonomous ODE design. To the best of our knowledge, we are the first to attempt using neural ODE to simultaneously address the action non-smoothness problem in deep RL caused by high-frequency input noise and large Lipschitz constant.

To address the aforementioned challenges, this study introduces the Smooth ODE (SmODE). Initially, the research presents a smooth ODE neuron designed to estimate action rate changes near the current state. Our theoretical proof demonstrates that this computation effectively controls the maximum state transition between adjacent temporal neurons. We then developed a SmODE neural network incorporating these smooth ODE neurons, which reduces action fluctuations by integrating additional regularization terms into the original policy objective. The primary goal of this network is to enhance control output smoothness and function as a versatile, plug-and-play policy approximator for a broad range of RL algorithms.

The key contributions of this paper are the following:

- We design a smooth ODE to function as a neuron of a NN for smooth control. This ODE neuron employs a mapping function to estimate the speed of change of the action in the neighborhood of the current state. Utilizing the estimated rate of change, it is possible to efficiently moderate the extent of neuronal hidden state alterations at contiguous time points, consequently reducing the difference in output from neighboring temporal neurons.

- The SmODE network is developed by utilizing the smooth ODE as neurons. Our network comprises three modules: the input module, the smooth ODE module, and the output module. The input module is a multi-layer perceptron (MLP) network and the output module is a linear transformation layer, with spectral normalization applied. The smooth ODE module consists of three layers, and the number of smooth ODE neurons in each layer can be selected according to the task complexity. This design endows the SmODE network with disturbance rejection and smoothness capabilities.

- We propose an SmODE-based RL algorithm designed to smooth action fluctuations. This algorithm incorporates the classical Actor-Critic architecture and integrates a SmODE network as its policy network. Our method reduces action fluctuations by combining two regularization terms with the original policy objective, aimed at augmenting state filtering

and controlling action fluctuation suppression in the SmODE network. In a three-degree-of-freedom vehicle trajectory tracking task, our approach achieves an 81.7% reduction in action fluctuation rate, while preserving performance, compared to using traditional MLPs as the policy network, under a Gaussian noise variance setting of 0.2.

Supplementary experimental outcomes confirm that the SmODE architecture surpasses MLPs and LipsNet in smoothing output while incurring negligible performance trade-offs. To accelerate adoption and further research, we have encapsulated SmODE as a PyTorch module, with the code available in the attached files.

Section 2 provides a simple introduction to online RL, a metric for measuring the ratio of action fluctuation in control outputs, and introduction to neural ODE. In Section 3, a new network architecture called SmODE is proposed, which includes smooth ODE neurons to smooth control outputs. The experimental results obtained from applying the proposed method are reported in Section 4. Section 5 provides the conclusions of this paper.

## 2 PRELIMINARIES

### 2.1 ONLINE REINFORCEMENT LEARNING

Standard RL settings involve discrete-time agent-environment interactions, typically modeled as continuous-state and continuous-action Markov Decision Processes (MDP) Sutton & Barto (2018). Feedback is provided through a bounded reward function $r(s_t, a_t)$, and state transitions are determined by the probability $p(s_{t+1}|s_t, a_t)$. State-action pairs are represented as $(s, a)$ for current and $(s', a')$ for subsequent. The agent's actions at state $s_t$ are guided by a stochastic policy $\pi(a_t|s_t)$, assigning probabilities to possible actions based on the current state.

In online RL, an agent learns and makes real-time decisions through interactions with its environment. A transition, $(s_t, a_t, r_t, s_{t+1})$, captures this interaction and is stored in an experience replay buffer, $\mathcal{R}$. During training, sampling from $\mathcal{R}$ produces data batches, promoting stable model training. The primary goal of online RL is to develop a policy that maximizes the expected cumulative return:

$$J_\pi = \mathbb{E}_{(s_{i \geq t}, a_{i \geq t}) \sim \pi}\Big[ \sum_{i=t}^{\infty} \gamma^{i-t} r(s_i, a_i) \Big], \tag{1}$$

where $\gamma \in (0, 1)$ represents the discount factor. The Q-value for a state-action pair $(s, a)$ is given by

$$Q(s, a) = \mathbb{E}_\pi\Big[ \sum_{i=0}^{\infty} \gamma^i r(s_i, a_i)|s_0 = s, a_0 = a \Big] \tag{2}$$

RL primarily uses an actor-critic architecture Li (2023), consisting of a policy function, $\pi$, and a corresponding Q-value function, $Q^\pi$. The policy iteration framework, used to derive the optimal policy $\pi^*$, alternates between policy evaluation and policy improvement. During policy evaluation, $Q^\pi$ is updated based on the self-consistency principle of the Bellman equation:

$$Q^\pi(s, a) = r(s, a) + \gamma \mathbb{E}_{s' \sim p, a' \sim \pi}[Q^\pi(s', a')]. \tag{3}$$

In the policy improvement phase, an enhanced policy $\pi_{\text{new}}$ is sought by optimizing current Q-value $Q^{\pi_{\text{old}}}$:

$$\pi_{\text{new}} = \arg\max_\pi \mathbb{E}_{s \sim d_\pi, a \sim \pi}[Q^{\pi_{\text{old}}}(s, a)]. \tag{4}$$

Practically, neural networks typically parameterize the policy and value functions, indicated as $\pi_\theta$ and $Q_\phi$. These functions are honed using gradient descent techniques to minimize the actor and critic loss functions, $\mathcal{L}_\pi(\theta)$ and $\mathcal{L}_q(\phi)$, respectively, which are formulated based on equation 4 and equation 3.

## 2.2 ACTION FLUCTUATION RATIO

To measure the action fluctuation of the control policy, Song et al. (2023) defined the action fluctuation ratio for continuous action settings:

$$\varepsilon(\pi) = \mathbb{E}_{\tau \sim \rho_\pi} \left[ \frac{1}{T} \sum_{t=1}^{T} \|a_t - a_{t-1}\| \right], \tag{5}$$

where $\rho_\pi$ is the state-action trajectory distribution induced by the policy $\pi$, $T$ is the episode length, $a_t$ and $a_{t-1}$ represent the action value at the current and previous time steps, respectively. It can be observed that the control smoothness is negatively correlated with the action fluctuation ratio $\varepsilon(\pi)$.

## 2.3 NEURAL ODE

Neural ODE treats the computation of NN as a process of solving ODE, enabling the model to efficiently handle continuous-time sequence problems and describe its dynamics through differential equation methods. Chen et al. (2018) proposed that the hidden state of a neural ODE can be defined by the solution of

$$\frac{\mathrm{d}x(t)}{\mathrm{d}t} = f\left(x(t), I(t), t, \theta\right), \tag{6}$$

where $x(t)$ represents the hidden states, $I(t)$ represents the input, $t$ represents time, $f$ is a NN with parameter $\theta$.

In control theory, a first-order low-pass filter Yuce & Minaei (2012) can be expressed in terms of an ODE as

$$\frac{\mathrm{d}x(t)}{\mathrm{d}t} = -\frac{x(t)}{\tau} + \frac{I(t)}{\tau}, \tag{7}$$

where $\tau$ is a time constant of the system. A larger $\tau$ value corresponds to a higher degree of filtering.

Instead of directly defining the derivatives of the hidden state using a neural network $f$, a more stable continuous-time recurrent neural network can be employed by the following equation Funahashi & Nakamura (1993):

$$\frac{\mathrm{d}x(t)}{\mathrm{d}t} = -\frac{x(t)}{\tau} + f\left(x(t), I(t), t, \theta\right). \tag{8}$$

Hasani et al. (2021) proposed the liquid time-constant (LTC), further explored the impact of the ODE structure on representation performance and proposed replacing $f\left(x(t), I(t), t, \theta\right)$ in equation 8 with $f\left(x(t), I(t), t, \theta\right)(A - x(t))$, where $A$ represents a learnable parameter.

Due to the reliance on advanced numerical ODE solvers, the training and inference speed of neural ODE is slow. This issue worsens as the complexity of the data, tasks, and state space increases. To address this, Hasani et al. (2022) derived a closed-form continuous-depth (CfC) model that preserves the modeling capabilities of ODE-based models without requiring a solver for data modeling.

## 3 SMOOTH ODE NETWORK

In this section, we first introduce the design of the ODE neuron. Then, we will describe the structure of the SmODE network in this paper. Following this, we propose an RL training approach devised to improve the smoothness of the policy while maintaining good control performance.

### 3.1 SMOOTH ORDINARY DIFFERENTIAL EQUATION

In order to address both the issue of high-frequency noise and the action non-smoothness caused by an unbounded Lipschitz constant, we have designed the ODE as follows.

To address the issue of high-frequency noise, we design the ODE with a low-pass structure, similar to equation 7. While the large time constant of the system ensures excellent action smoothness, it also introduces additional delay. These delays can significantly harm control performance when the system needs a fast response. To address this issue, we introduce a learnable function $f(x(t), I(t), t, \theta)$ that

maps the input signal $I(t)$ and the neuronal hidden state $x(t)$ to the inverse of the time constant $\frac{1}{\tau}$. The equation is shown as

$$\frac{\mathrm{d}x(t)}{\mathrm{d}t} = -f\left(x(t), I(t), t, \theta\right) x(t) + f\left(x(t), I(t), t, \theta\right) I(t), \tag{9}$$

where $f$ is a NN with parameter $\theta$. Since the time constant must be a positive number, the function $f$ must be greater than 0.

The magnitude of the Lipschitz constant can be controlled by constraining the size of $|\frac{\mathrm{d}x(t)}{\mathrm{d}t}|$, and this constraint must be state-dependent; otherwise, it may impair performance in regions where certain systems require a faster response.

In this paper, we replace $I(t)$ on the far right side of equation 9 with a learnable function $g(x(t), I(t), t, \theta)$, resulting in the following equation:

$$\frac{\mathrm{d}x(t)}{\mathrm{d}t} = -f\left(x(t), I(t), t, \theta\right) x(t) + f\left(x(t), I(t), t, \theta\right) g(x(t), I(t), t, \theta). \tag{10}$$

Based on equation 10, we can draw the following theorem:

**Theorem 1.** *Let $x_i$ denote the hidden state of a neuron $i$ within the smooth ODE, identified by equation 10, and let neuron $i$ receive some incoming connections. Then, the hidden state of any neuron $i$, on a finite interval $Int \in [0, T]$, is bounded as follows:*

$$\min(0, g(x(t), I(t), t, \theta)_i^{\min}) \leq x_i(t) \leq \max(0, g(x(t), I(t), t, \theta)_i^{\max}). \tag{11}$$

*Proof.* See Appendix A.1.

Theorem 1 suggests that $g\left(x(t), I(t), t, \theta\right)$, which we designed, guarantees that the hidden state of a neuron remains bounded by equation 11 for a finite time. Additionally, $g\left(x(t), I(t), t, \theta\right)$ is state-dependent, allowing for the adaptive adjustment of the hidden state boundaries of neurons based on the current state. This conclusion is further supported by the analytical results presented in Appendix D.

Using a bionic modeling method similar to that in Lechner et al. (2020), we can obtain the specific formulation of our smooth ODE neuron, which is presented as follows:

$$\frac{\mathrm{d}x_i}{\mathrm{d}t} = \sum_j \left[ -\frac{w_{ij}}{C_{\mathrm{m}_i}} \sigma_i\left(x_j\right) x_i + \frac{w_{ij}}{C_{\mathrm{m}_i}} \sigma_i\left(x_j\right) \cdot \tanh(h\left(x_j, \theta\right)) \right] + x_{\mathrm{leak}_i}, \tag{12}$$

where $w_{ij} \in (0.001, 1.0)$ denotes the synaptic weight from neuron $i$ to neuron $j$, and $C_{\mathrm{m}_i} \in (0.4, 0.6)$ signifies the membrane capacitance. The term $x_{\mathrm{leak}_i}$ refers to the resting potential of a neuron. The sigmoid function $\sigma_i\left(x_j\right) = \frac{1}{1+e^{-\gamma_{ij}(x_j - \mu_{ij})}}$ is introduced, where $\gamma_{ij}$ and $\mu_{ij}$ are trainable parameters with initial values ranging from 3 to 8 and 0.3 to 0.8, respectively. Furthermore, $f(x(t), I(t), t, \theta)$ is expressed as $\frac{w_{ij}}{C_{\mathrm{m}_i}} \sigma_i\left(x_j\right)$, and $I(t)$ is equal to $x_j$. $g\left(x(t), I(t), t, \theta\right)$ is equal to $\tanh(h(x_j, \theta))$, representing a NN.

Based on equation 10, equation 11 and equation 12 , we can also obtain the following theorem:

**Theorem 2.** *Let $x_i$ denote the hidden state of a neuron $i$ within the smooth ODE, identified by equation 10. Then, the absolute value of the derivative of the hidden state concerning time for any neuron $i$ has an upper bound controlled by $M(x(t), I(t), t, \theta)_i$, as follows*

$$|\frac{\mathrm{d}x_i(t)}{\mathrm{d}t}| \leq M(x(t), I(t), t, \theta)_i \cdot C, \tag{13}$$

*where $\max(|g(x(t), I(t), t, \theta)_i^{\min}|, |g(x(t), I(t), t, \theta)_i^{\max}|) = M(x(t), I(t), t, \theta)_i$, $C$ is a bounded positive constant.*

*Proof.* See Appendix A.2.

Theorem 2 shows that the hidden state of a smooth ODE neuron has an upper bound on the absolute value of the temporal derivative controlled by $M(x(t), I(t), t, \theta)_i$. Therefore, we can suppress the value of $|\frac{\mathrm{d}x_i(t)}{\mathrm{d}t}|$ by suppressing the value of $M(x(t), I(t), t, \theta)_i$.

The nonlinear characteristics of semantics present challenges in deriving an analytical solution for equation 12. As a result, we opt for a numerical ODE solver. To strike a balance between computational efficiency, solution accuracy, and stability, we select the fixed time-step semi-implicit Euler discretization method Ethier & Bourgault (2008) to solve this equation. We can unroll a given dynamical system of the form $\frac{\mathrm{d}x(t)}{\mathrm{d}t} = l(x(t), x(t + \Delta t))$ by

$$x(t + \Delta t) = x(t) + \Delta t \cdot l(x(t), x(t + \Delta t)). \tag{14}$$

Applying the fixed time-step semi-implicit Euler discretization method to equation 12, we can obtain

$$
\begin{aligned}
x_i(t + \Delta t) = {} & \frac{x_i(t)\frac{C_{\mathrm{m}_i}}{\Delta t} + C_{\mathrm{m}_i} x_{\mathrm{leak}_i}}{\frac{C_{\mathrm{m}_i}}{\Delta t} + C_{\mathrm{m}_i} + \sum_{j \epsilon I_{\mathrm{in}}} w_{ij} \sigma_i(x_j(t))} \\
& + \frac{\sum_{j \epsilon I_{\mathrm{in}}} w_{ij} \sigma_i(x_j(t)) \cdot \tanh(h(x_j(t), \theta))}{\frac{C_{\mathrm{m}_i}}{\Delta t} + C_{\mathrm{m}_i} + \sum_{j \epsilon I_{\mathrm{in}}} w_{ij} \sigma_i(x_j(t))},
\end{aligned}
\tag{15}
$$

where $I_{\mathrm{in}}$ represents the set of neurons that have connections to neuron $i$. During the training phase of solving ODE, we initialize the hidden states uniformly to zero. During the sampling phase of solving ODE, the hidden state is initially set to zero in the first sampling step, followed by using the hidden state value from the preceding step for subsequent initialization. In this study, the numerical ODE solver has an iteration step size of 6 and a discrete interval time of 1.

## 3.2 THE SMODE NETWORK ARCHITECTURE

To improve the smoothness of control outputs, we further introduce the SmODE network, employing the smooth ODE as its neuron. It is applicable as a policy network across a wide range of RL frameworks. The architecture of the SmODE is shown in Fig. 1, which is structured with an input module, a smooth ODE module, and an output module. The input module is a MLP network, and the output module is a linear transformation layer, with spectral normalization applied. The smooth ODE module consists of three layers, and the number of smooth ODE neurons in each layer can be selected according to the task complexity.

## 3.3 SMODE-BASED RL

To facilitate smoother control, the SmODE network is utilized for parameterizing the actor in RL, represented by $\pi_\theta$, where $\theta$ denotes the respective network parameters. MLP is still utilized for parameterizing critic in RL, represented by $Q_\phi$.

The magnitude of the $f(x(t), I(t), t, \theta)$ value indicates the extent of filtering; a smaller value corresponds to a higher degree of filtering, thereby more effectively suppressing high-frequency noise interference. Therefore, we add the coefficient $f(x(t), I(t), t, \theta)$ associated with the filtering as a regularization term. The function $\tanh(h(x(t), I(t), t, \theta))$ regulates the range of values for the hidden state of the neuron; a smaller absolute value of $h(x(t), I(t), t, \theta)$ results in more pronounced inhibition of the magnitude change of hidden state across neighboring time steps. Therefore, the coefficient $h^2(x(t), I(t), t, \theta)$ is associated with the hidden state boundary value as a regularization term. Both regular terms are added to the original RL training loss. The modified actor loss is

$$\min \mathcal{L}'_\pi(\theta) = \mathcal{L}_\pi(\theta) + \lambda_1 \mathbb{E}_{s \sim \mathcal{R}} \left[ \sum_{i=0}^{N} f(\cdot) \right] + \lambda_2 \mathbb{E}_{s \sim \mathcal{R}} \left[ \sum_{i=0}^{N} h^2(\cdot) \right], \tag{16}$$

where $\lambda_1$ and $\lambda_2$ are the regularization factors, $\mathcal{R}$ is the replay buffer, the first regularization term is named the time constant term, the second regularization term is named state boundary term, and $N$ is the number of smooth ODE neurons of the SmODE network. The pseudocode of SmODE-based RL is illustrated in Algorithm 1.

## 4 EXPERIMENTS

## 4.1 EXPERIMENTAL ENVIRONMENT

In this study, ten types of experimental environments are adopted to validate the efficacy of the SmODE network: a vehicle trajectory tracking task, a linear quadratic regulator problem, and eight

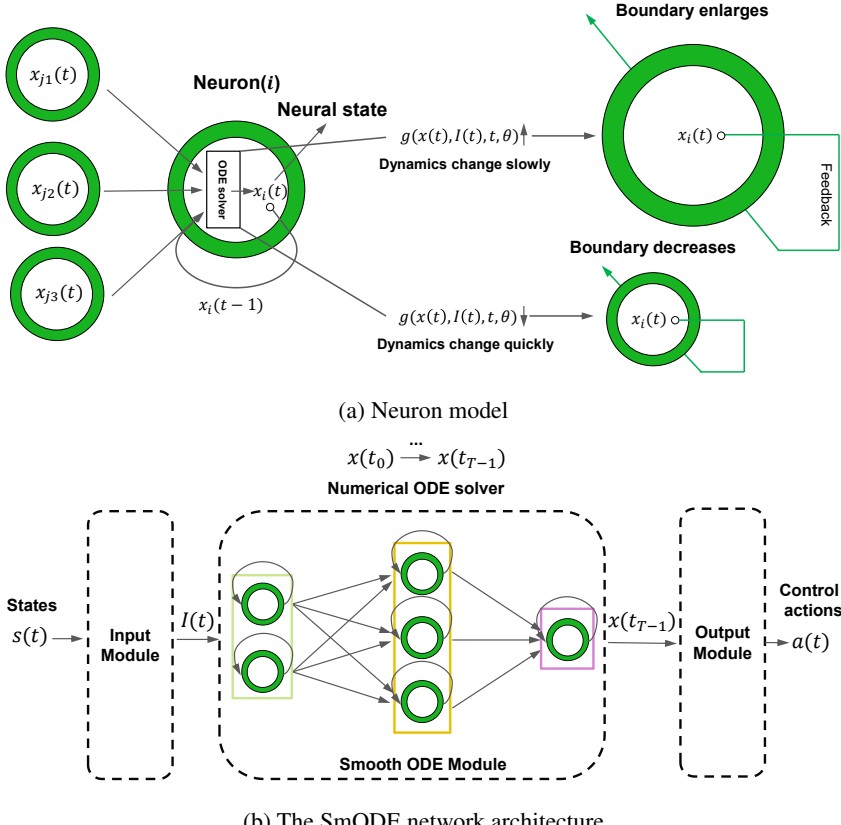

Figure 1: **Designing SmODE network with smooth ODE neuron.** (a) The neural state, $x_i(t)$, of a smooth ODE neuron $i$ integrates inputs from neurons $j1, j2, j3$ and its previous state. The system dynamics, $g(x(t), I(t), t, \theta)$, allow for adaptive state boundary adjustments during the solving process by the numerical ODE solver, akin to a feedback control mechanism. (b) The SmODE network consists of an input module, a smooth ODE module, and an output module. The input module is a MLP network, while the output module features a linear transformation layer with spectral normalization applied. The smooth ODE module contains three layers, with the number of neurons in each layer tailored to the task's complexity. $T$ denotes the number of iterations performed by the numerical ODE solver.

robotic control tasks in Mujoco Todorov et al. (2012). All experiments were conducted on eight AMD Ryzen Threadripper 3960X 24-core processors with 128G of RAM each. The time required for Mujoco tasks with an average training step length of 1 million is 14h.

Vehicle trajectory tracking is a significant problem in autonomous driving. We simulated the motion of the vehicle using the vehicle dynamics model proposed by Ge et al. (2021). Furthermore, we chose an LQR problem with two states and one action as an ablation experiment task. Detailed introductions to the two experimental environments are provided in the Appendix C.

Mujoco is a benchmark RL environment that integrates several robot control tasks. The specific simulation tasks, depicted in Fig. 4, include Humanoid, Pusher, Hopper, Reacher, Walker2d, Ant, InvertedDoublePendulum and CarRacing.

We will use the following two types of RL algorithms. Infinite-time approximate dynamic programming (INFADP)Li (2023) is a typical model-based RL algorithm. Distributional soft actor-critic (DSAC)Duan et al. (2021) is a typical model-free RL algorithm. All experiments were conducted in general optimal control problem solver (GOPS)Wang et al. (2023), and the results are averaged over five random seeds.

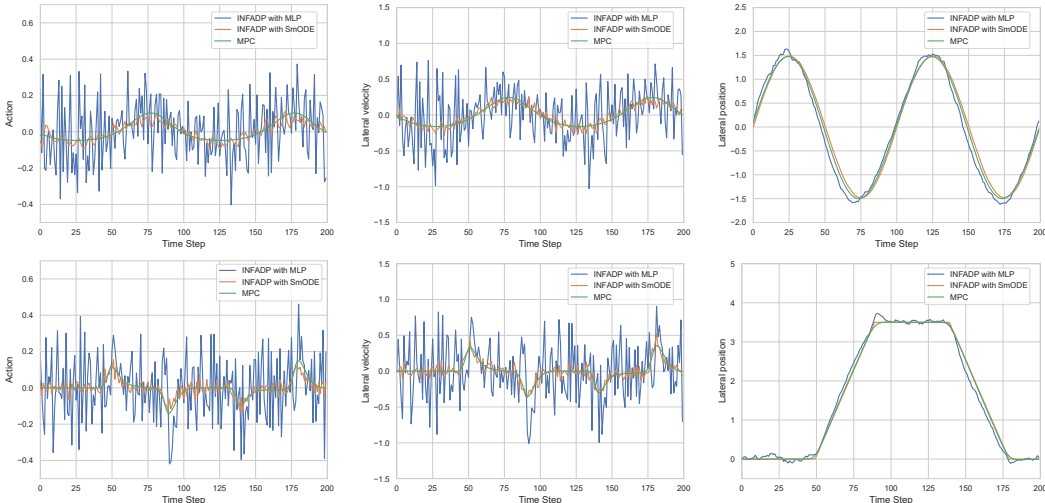

Figure 2: **Results in vehicle trajectory tracking environment.** In this experiment, MPC operates without adding noise, and its control outcomes will serve as a benchmark for the optimal policy. On the first line is the result of the sine curve, and on the second line is the result of the double lane-change curve.

## 4.2 TEST RESULTS ON VEHICLE TRACKING PROBLEM

Table 1: Performance analysis of tracking a double lane-change curve using the INFADP algorithm. Results are expressed as mean ± standard deviation of five independent environmental seeds.

| Policy network | Gaussian noise standard deviation | | | | | | | |
|---|---|---|---|---|---|---|---|---|
| | 0.05 | | 0.10 | | 0.15 | | 0.20 | |
| | $\varepsilon(\pi)$ | TAR | $\varepsilon(\pi)$ | TAR | $\varepsilon(\pi)$ | TAR | $\varepsilon(\pi)$ | TAR |
| SmODE | **0.061±0.011** | **-0.778±0.022** | **0.110±0.013** | **-0.836±0.026** | **0.137±0.018** | **-0.876±0.032** | **0.176±0.022** | **-0.926±0.028** |
| MLP | 0.312±0.022 | -1.111±0.038 | 0.645±0.038 | -1.725±0.049 | 0.842±0.047 | -2.266±0.050 | 0.964±0.059 | -3.087±0.047 |
| LTC | 0.193±0.015 | -0.920±0.027 | 0.392±0.020 | -1.281±0.031 | 0.589±0.021 | -1.848±0.046 | 0.778±0.034 | -2.334±0.036 |
| LipsNet | 0.069±0.013 | -0.893±0.018 | 0.125±0.019 | -0.934±0.020 | 0.169±0.019 | -0.987±0.021 | 0.221±0.027 | -1.112±0.033 |
| MPC | 0.238±0.008 | -0.818±0.013 | 0.439±0.010 | -1.248±0.017 | 0.554±0.015 | -1.509±0.028 | 0.706±0.022 | -2.252±0.026 |

We illustrate our approach by tracking the double lane-change curve, employing five distinct methods: INFADP with MLP, INFADP with SmODE, INFADP with LipsNet, INFADP with LTC and MPC Holkar & Waghmare (2010). Table 1 displays the performance metrics for these methods, noting that Gaussian noise is the noise type used. In this context, TAR denotes the total average return, and $\varepsilon(\pi)$ represents the action fluctuation ratio.

In four distinct noise environments with varying levels, our algorithm consistently outperformed others. As shown in the table, SmODE, acting as a policy network, significantly reduces the action fluctuation ratio and enhances the TAR compared to MLP. Notably, with a Gaussian noise variance of 0.2, our network lowers the action fluctuation ratio by about 81.7%, demonstrating superior smoothness. The results for the first and third algorithms indicate that our neural network architecture, combined with modified actor loss, greatly mitigates the action fluctuation rate. Moreover, in noisy environments, our approach exceeds the recent LipsNet enhancement in performance and proves more effective than the classical MPC controller.

This resilience is largely due to the low-pass filtering effect and SmODE's ability to suppress its Lipschitz constant. Given the common presence of noise in real-world settings, SmODE's robustness in noisy environments is of significant practical value.

Furthermore, with a Gaussian noise variance of 0.05, our analysis of experimental results using MLP and SmODE as policy networks for tracking sine and double lane-change curves shows notable differences. We used the MPC algorithm as a baseline for noise-free comparison. As illustrated in Fig. 2, SmODE not only exhibits a lower action fluctuation ratio than MLP but also smaller variations in lateral velocity, enhancing vehicle comfort and safety.

## 4.3 TEST RESULTS ON MUJOCO BENCHMARK

In this experiment, we focused on eight robotic control tasks within the Mujoco environment. We employed DSAC Duan et al. (2021) as the fundamental RL algorithm, configuring the policy networks as MLP, LipsNet, LTC, and SmODE. The assessment was performed under two levels of Gaussian noise to mimic various real-world conditions. Since the state values of different Mujoco tasks vary greatly, we set two levels of Gaussian noise for the eight tasks, as shown in Table 2.

Table 2: Variance of different levels of Gaussian noise for different Mujoco tasks.

| Noise level | InvertedDoublePendulum-v3 | Reacher-v2 | Humanoid-v3 | Pusher-v2 | Hopper-v3 | Walker2d-v3 | Ant-v3 | CarRacing-v1 |
|---|---|---|---|---|---|---|---|---|
| level 1 | 0.005 | 0.050 | 0.020 | 0.050 | 0.050 | 0.050 | 0.050 | 0.150 |
| level 2 | 0.015 | 0.100 | 0.050 | 0.100 | 0.100 | 0.100 | 0.070 | 0.250 |

For the whole task, noise is added to all states. The results, which are the averages of five seeds over 1 million training steps, are shown in Table 3.

Under different levels of Gaussian noise, SmODE, functioning as a policy network, achieved the lowest average action fluctuations compared to LTC, LipsNet, and MLP. Additionally, SmODE exhibited the best performance in most Mujoco tasks. Given that the pursuit of action smoothness and high performance can be somewhat contradictory, it is understandable that the best performance was not achieved in all experimental settings. Moreover, we also experimented with TD3 Fujimoto et al. (2018) in the Walker2d-v3 and Ant-v3 environments and obtained similar results, as shown in Appendix E.

Table 3: Average control performance of SmODE, LTC, LipsNet, and MLP for different Gaussian noise levels, where level 1 is on the left column and level 2 is on the right column. The average action fluctuation rate is indicated in parentheses. Results are expressed as mean ± standard deviation of five independent environmental seeds.

| Network structure | InvertedDoublePendulum-v3 | | Reacher-v2 | |
|---|---|---|---|---|
| SmODE | **9357±2 (0.15)** | **9340±2 (0.44)** | **-5.67±1 (0.22)** | **-9.16±1 (0.32)** |
| LTC | 9355±2 (0.25) | 9336±3 (0.64) | -6.09±2 (0.31) | -10.29±3 (0.42) |
| LipsNet | 9357±2 (0.20) | 9338±2 (0.50) | -5.94±1 (0.26) | -9.84±2 (0.39) |
| MLP | 9357±2 (0.27) | 9335±4 (0.68) | -5.73±3 (0.30) | -10.49±3 (0.44) |

| Network structure | Humanoid-v3 | | Pusher-v2 | | Hopper-v3 | |
|---|---|---|---|---|---|---|
| SmODE | 10819±81 (**0.45**) | **10746±101 (0.50)** | **-40±1 (0.90)** | **-51±1 (1.39)** | **3265±232 (0.70)** | **2532±302 (1.01)** |
| LTC | 10626±128 (0.60) | 10578±245 (0.66) | -44±2 (1.51) | -86±8 (2.30) | 2724±287 (0.88) | 1398±345 (1.25) |
| LipsNet | 10872±89 (0.57) | 10715±104 (0.62) | -43±2 (1.23) | -55±3 (2.03) | 2905±301 (0.84) | 1787±291 (1.21) |
| MLP | **10892±342** (0.62) | 10567±512 (0.69) | -49±3 (2.01) | -71±3 (2.60) | 1282±322 (0.93) | 1108±231 (1.30) |

| Network structure | Walker2d-v3 | | Ant-v3 | | CarRacing-v1 | |
|---|---|---|---|---|---|---|
| SmODE | **6039±112 (0.94)** | **5037±114 (1.30)** | 3564±184 (**1.68**) | **1677±41 (1.93)** | **916±27 (0.83)** | **873±14 (0.96)** |
| LTC | 5861±482 (1.10) | 2352±604 (1.71) | 2872±341 (2.04) | 1084±298 (2.16) | 906±21 (0.87) | 694±116 (1.03) |
| LipsNet | 6032±238 (1.05) | 4981±423 (1.45) | **3721±212** (1.93) | 1532±109 (2.10) | 896±31 (0.85) | 821±56 (1.00) |
| MLP | 5663±508 (1.21) | 1597±815 (1.80) | 1086±1246 (2.16) | 197±120 (2.30) | 870±38 (0.88) | 751±86 (1.05) |

## 4.4 ABLATION STUDY

To demonstrate how the time constant and state boundary regularization terms contribute to the final smoothing action, we conducted ablation experiments. We still use the model-based RL method, INFADP Li (2023), to train in this environment. All ablation experiments are performed on the linear quadratic regulation problem for two-dimensional states and one-dimensional action.

The following are the specific designs of three ablation experiments: 1) **SmODE w/o time constant term: ablation time constant regular term.** This ablation experiment aimed to validate the impact of incorporating the time constant of the system as a regular term in actor loss on the smoothing of action output. 2) **SmODE w/o state boundary term: ablation regular term for the boundary of neuron states.** This ablation experiment involved removing the regular term from the actor loss, a term that adaptively controls neuron state boundaries by predicting the rate of change in actions near the current state. 3) **Baseline: ablation the both regular terms.** This ablation experiment simultaneously removes the two regular terms added to the actor loss and replaces the MLP with a neural ODE network only.

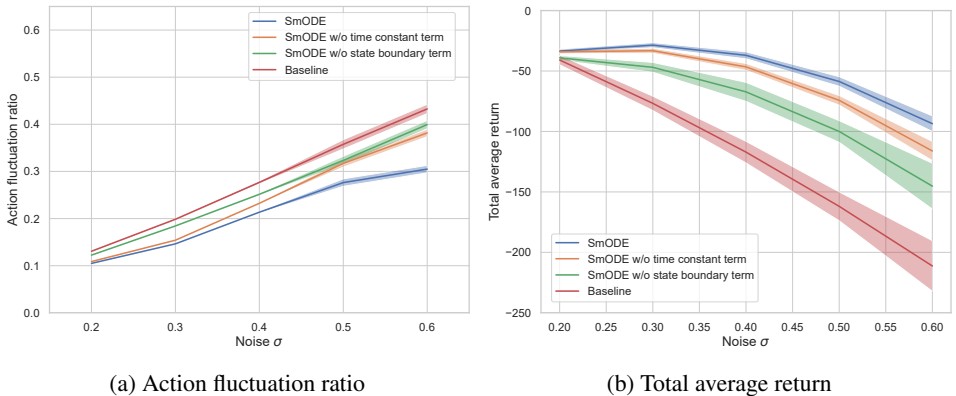

(a) Action fluctuation ratio  (b) Total average return

Figure 3: **Results in the LQR environment.** The X-axis is the noise variance.

In Fig. 3, we present results from three ablation studies in the LQR environment using the INFADP algorithm. We introduced various levels of uniform noise into the observed state. The results show that compared to the baseline, both regularization terms effectively reduce the action fluctuation ratio and increase the total average reward, with the state boundary regularization term having a particularly notable impact. Notably, with a noise variance of 0.6, SmODE decreases the action fluctuation rate by 12% and boosts the total average return by 79%. As noise levels increase, SmODE shows a slower rise in action fluctuation ratio and a more gradual decrease in total average return compared to the baseline neural ODE network, highlighting its superior noise resistance and smoothing capabilities.

In addition, we conducted ablation experiments on whether the output module used spectral normalization (SN) techniques in the Walker2d and Humanoid tasks, with the experimental results shown in Table 4.

Table 4: Control performance of different network structures under different Gaussian noisy variance. The average action fluctuation rate is indicated in parentheses.

| Network structure | Walker2d-v3 | | Humanoid-v3 | |
|---|---|---|---|---|
| SmODE | 6039±112 (**0.73**) | 5037±114 (**1.03**) | 10819±81 (**0.45**) | 10746±101 (**0.50**) |
| SmODE-wo-SN | 6013±202 (0.80) | 5165±164 (1.12) | 10821±63 (0.48) | 10739±122 (0.54) |

The experimental results indicate that using SN techniques can further reduce action fluctuation, with minimal impact on overall performance. It is worth noting that the reduction in action fluctuation due to the use of SN techniques is relatively small compared to the overall decrease.

## 5  CONCLUSION

In this study, we introduce the SmODE network to tackle non-smooth action outputs in deep reinforcement learning. The network features a smooth ODE as a key component of its neurons, enabling adaptive state boundary adjustments and low-pass filtering. This design grants the neurons disturbance rejection and smoothness capabilities. As a policy network, SmODE enhances control output smoothness and increases average rewards in various RL algorithms over MLP and LipsNet. We hope our contributions advance real-world RL applications.

## 6  LIMITATION

Solving the Neural ODE using numerical methods requires $N$ iterations. In our case, we balanced solution accuracy and computational efficiency by adopting $N = 6$, a value commonly used in previous related work. As a result, the training time for SmODE increases by a factor of 2 to 3 compared to MLP, which is an issue that needs to be addressed. In future work, we plan to explore the latest techniques for training neural ODE networks to accelerate the backpropagation process.

# 7 ACKNOWLEDGEMENTS

This study is supported by National Key R&D Program of China with 2022YFB2502901. It is also partially supported by Tsinghua University-Toyota Joint Research Center for AI Technology of Automated Vehicle, and supported by Tsinghua University Initiative Scientific Research Program.

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

# A  THEORETICAL RESULTS

## A.1  PROOF OF THEOREM 1

**Theorem 1** Let $x_i$ denote the hidden state of a neuron $i$ within the smooth ODE, identified by equation 10, and let neuron $i$ receive some incoming connections. Then, the hidden state of any neuron $i$, on a finite interval $Int \in [0, T]$, is bounded as follows:

$$\min(0, g(x(t), I(t), t, \theta)_i^{\min}) \le x_i(t) \le \max(0, g(x(t), I(t), t, \theta)_i^{\max}). \tag{17}$$

*Proof.* Let us insert $M = \max(0, g(\cdot)_i^{\max})$ as the neural state of neuron $i$, $x_i(t)$ into equation 10:

$$\frac{dx_i}{dt} = \underbrace{- f(\mathbf{x}_j(t), t, \theta)M + f(\mathbf{x}_j(t), t, \theta) \cdot g(\mathbf{x}_j(t), t, \theta)_i}_{\le 0}. \tag{18}$$

The right-hand side of equation 18 is negative, considering the constraints on $M$, the positivity of weights, and the fact that $f(x_j)$ is positive. Consequently, the left-hand side must also be negative. Employing an approximation on the derivative term yields the following relationship:

$$\frac{dx_i}{dt} \le 0, \quad \frac{dx_i}{dt} \approx \frac{x_i(t + \Delta t) - x_i(t)}{\Delta t} \le 0. \tag{19}$$

By substituting $x_i(t)$ with $M$, we get:

$$\frac{x(t + \Delta t) - M}{\Delta t} \le 0 \rightarrow x(t + \Delta t) \le M, \tag{20}$$

which means $x_i(t) \le \max(0, g(\cdot)_i^{\max})$. We can also obtain similar results $\min(0, g(\cdot)_i^{\min}) \le x_i(t)$.

## A.2  PROOF OF THEOREM 2

**Theorem 2** Let $x_i$ denote the hidden state of a neuron $i$ within the smooth ODE, identified by equation 10. Then, the absolute value of the derivative of the hidden state concerning time for any neuron $i$ has an upper bound controlled by $M(x(t), I(t), t, \theta)_i$, as follows

$$|\frac{\mathrm{d}x_i(t)}{\mathrm{d}t}| \le M(x(t), I(t), t, \theta)_i \cdot C \tag{21}$$

where $\max(|g(x(t), I(t), t, \theta)_i^{\min}|, |g(x(t), I(t), t, \theta)_i^{\max}|) = M(x(t), I(t), t, \theta)_i$, $C$ is a bounded positive constant.

*Proof.*

$$\frac{\mathrm{d}x(t)}{\mathrm{d}t} = -f(x(t), I(t), t, \theta)x(t) + f(x(t), I(t), t, \theta) \, g(x(t), I(t), t, \theta)$$

$$\Rightarrow |\frac{\mathrm{d}x(t)}{\mathrm{d}t}| = |-f(x(t), I(t), t, \theta)x(t) + f(x(t), I(t), t, \theta) \, g(x(t), I(t), t, \theta)|$$

$$\le |f(x(t), I(t), t, \theta)x(t)| + |f(x(t), I(t), t, \theta) \, g(x(t), I(t), t, \theta)|$$

$$\le |f(x(t), I(t), t, \theta)| \cdot |x(t)| + |f(x(t), I(t), t, \theta) \, g(x(t), I(t), t, \theta)|$$

$$\le f(x(t), I(t), t, \theta) \cdot M(x(t), I(t), t, \theta) + f(x(t), I(t), t, \theta) \cdot M(x(t), I(t), t, \theta) \quad \text{According to equation 17}$$

$$= M(x(t), I(t), t, \theta) \cdot 2f(x(t), I(t), t, \theta)$$

$$\le M(x(t), I(t), t, \theta) \cdot C$$

where $f(x(t), I(t), t, \theta) = \frac{w_{ij}}{C_{m_i}}\text{sigmoid}(\cdot)$, $w_{ij} \in (0.001, 1.0), C_{m_i} \in (0.4, 0.6)$, $C$ is s a bounded positive constant.

The output module is a simple layer of linear mappings $a = wx + b$, with spectral normalization applied, so there $|\frac{\mathrm{d}a(t)}{\mathrm{d}t}| \propto |\frac{\mathrm{d}x(t)}{\mathrm{d}t}|$ holds.

$$\Rightarrow |\frac{da(t)}{dt}| \le M(x(t), I(t), t, \theta) \cdot C'$$

where $C'$ is a bounded positive constant.

## B    PSEUDOCODE

---
**Algorithm 1** Training method of SmODE-based RL.
---
Input: $\theta, \phi, \lambda_1, \lambda_2, \beta_q, \beta_\pi$
**for** each iteration **do**
    Collect a batch of samples $(s, a, r, s')$ with policy $\pi_\theta$
    Store the samples in replay buffer $\mathcal{R}$
    **for** each update step **do**
        Sample data from $\mathcal{R}$
        Update actor using $\theta \leftarrow \theta - \beta_\pi \nabla_\theta \mathcal{L}'_\pi(\theta)$
        Update critic using $\phi \leftarrow \phi - \beta_q \nabla_\phi \mathcal{L}_q(\phi)$
    **end for**
**end for**
---

## C    EXPERIMENTAL ENVIRONMENT INTRODUCTION

### C.1    VEHICLE TRAJECTORY TRACKING ENVIRONMENT

Table 5 provides detailed descriptions of the states and actions in the vehicle trajectory tracking task.

Table 5: List of states and actions

| Varible | | Description | Uint |
|---|---|---|---|
| State | $x$ | longitudinal position | m |
| | $y$ | lateral position | m |
| | $\varphi$ | heading angle | rad |
| | $u$ | longitudinal velocity | m/s |
| | $v$ | lateral velocity | m/s |
| | $\omega$ | yaw rate at center of gravity (C.G.) | rad/s |
| Action | $a$ | longitudinal acceleration | m/s$^2$ |
| | $\delta$ | front wheel angle | rad |

In the vehicle trajectory tracking experiment, we selected sine and double lane-change curves for tracking.

The reward is designed as

$$\begin{aligned} r = &-0.04 \left(x - x_{\text{ref}}\right)^2 - 0.04 \left(y - y_{\text{ref}}\right)^2 \\ &-0.02 \left(\varphi - \varphi_{\text{ref}}\right)^2 - 0.02 \left(u - u_{\text{ref}}\right)^2 \\ &-0.01\omega^2 - 0.01\delta^2 - 0.01a^2, \end{aligned} \tag{22}$$

where $x_{\text{ref}}, y_{\text{ref}}, \varphi_{\text{ref}}, u_{\text{ref}}$ represent reference states.

The vehicular parameters are listed in Table 6, where C.G. means the center of gravity.

Table 6: Vehicular parameters

| Parameter | Description | Value |
|:---:|:---:|:---:|
| $m$ | mass of the vehicle | 1412 kg |
| $l_f$ | distance between C.G. and front axle | 1.06 m |
| $l_r$ | distance between C.G. and rear axle | 1.85 m |
| $k_f$ | front axle equivalent sideslip stiffness | -128916 N/rad |
| $k_r$ | rear axle equivalent sideslip stiffness | -85944 N/rad |
| $I_z$ | yaw inertia of vehicle body | 1536.7 kg·m$^2$ |
| $f$ | control frequency | 10 Hz |

## C.2 LINEAR QUADRATIC REGULATION PROBLEM

The state-space equation is

$$\dot{X} = AX + BU, \tag{23}$$

where

$$A = \begin{bmatrix} 0 & 1 \\ 0 & 0 \end{bmatrix}, \ B = \begin{bmatrix} 0 \\ 1 \end{bmatrix}. \tag{24}$$

The reward is designed as

$$r_t = -X_t^{\mathrm{T}} Q X_t - U_t^{\mathrm{T}} R U_t, \tag{25}$$

where

$$Q = diag(2, 1), \ R = 1, \tag{26}$$

$diag$ means a diagonal matrix.

## C.3 MUJOCO ENVIRONMENTS

Mujoco serves as a benchmark RL environment comprising various robot control tasks. The specific simulation tasks, shown in Fig. 4, include Humanoid, Pusher, Hopper, Reacher, Walker2d, Ant, Inverted Double Pendulum, and Car Racing.

## D LANDSCAPE OF AVERAGE $|h(x_j, \theta)|$

In the LQR problem used for the ablation experiments, we plot the average values of $|h(x_j, \theta)|$ in SmODE, as shown in Fig. 5. Notably, $|h(x_j, \theta)|$ exhibits larger values around the states $(-1, -1)$ and $(1, 1)$, as these states indicate a departure from the steady state $(0, 0)$. Consequently, the Lipschitz constant may be larger, necessitating an increase in the range of values for the hidden state of the neuron.

## E TD3 WITH SMODE

In order to demonstrate the smoothing ability of SmODE in other RL algorithms, we experimented with TD3 as an example in Walker2d-v3 and Ant-v3 environments, as shown in Table 7.

Table 7: Control performance of different network structures under Gaussian noisy variance of 0.05 (left column) and Gaussian noisy variance of 0.1 (right column) conditions. The average action fluctuation rate is indicated in parentheses.

| Network structure | Walker2d-v3 | | Ant-v3 | |
|:---:|:---:|:---:|:---:|:---:|
| SmODE | **3962±361 (0.87)** | **3504±773 (1.11)** | **4158±524 (1.01)** | **3857±754 (1.67)** |
| LipsNet | 3578±392 (0.95) | 3226±623 (1.32) | 4002±531 (1.24) | 3398±482 (1.88) |
| MLP | 3226±360 (1.08) | 2063±520 (1.60) | 3852±227 (1.72) | 862±242 (2.19) |

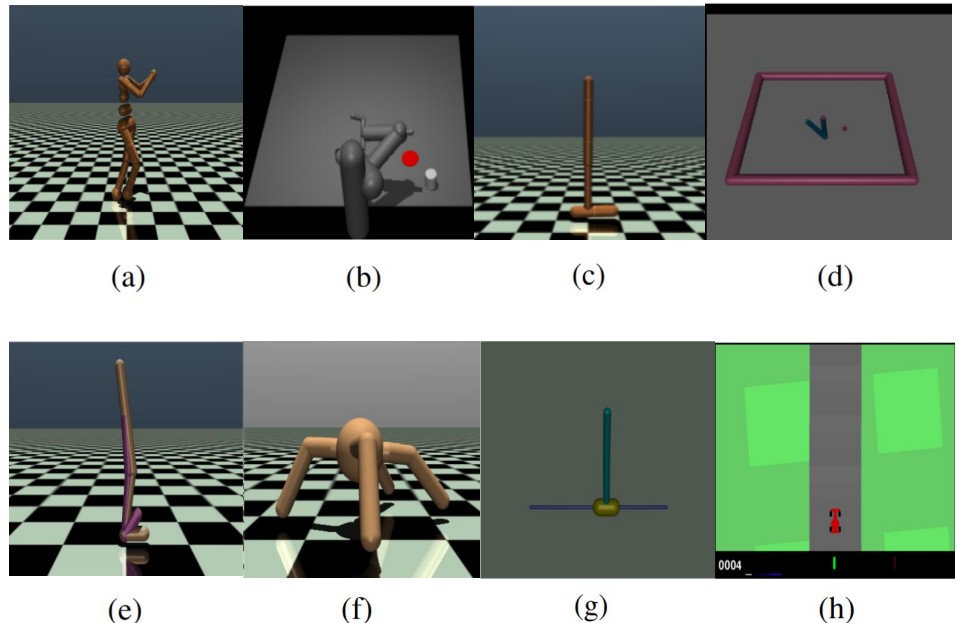

Figure 4: **Simulation tasks.** (a) Humanoid-v3:$(s \times a) \in \mathbb{R}^{376} \times \mathbb{R}^{17}$. (b) Pusher-v2: $(s \times a) \in \mathbb{R}^{23} \times \mathbb{R}^7$. (c) Hopper-v3 : $(s \times a) \in \mathbb{R}^{11} \times \mathbb{R}^3$. (d) Reacher-v2: $(s \times a) \in \mathbb{R}^{11} \times \mathbb{R}^2$. (e) Walker2d-v3: $(s \times a) \in \mathbb{R}^{17} \times \mathbb{R}^6$. (f) Ant-v3: $(s \times a) \in \mathbb{R}^{111} \times \mathbb{R}^8$. (g) InvertedDoublePendulum-v3: $(s \times a) \in \mathbb{R}^{11} \times \mathbb{R}^1$. (h) CarRacing-v1: $(s \times a) \in \mathbb{R}^{96 \times 96 \times 3} \times \mathbb{R}^2$ (image-input).

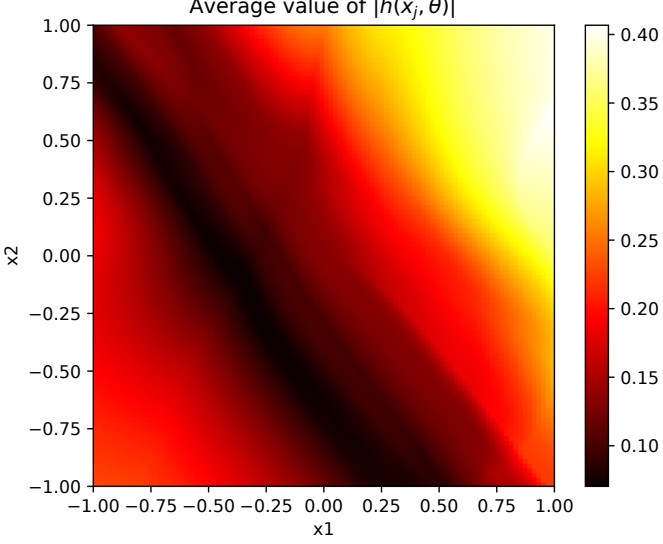

Figure 5: Landscape of average $|h(x_j, \theta)|$.

## F  TRAINING DETAILS

In Mujoco tasks, hyperparameters unrelated to SmODE were consistent with those in the DSAC paper. The parameters that needed adjustment were only $\lambda_1$ and $\lambda_2$, as well as the number of neurons in the three-layer network of the smooth ODE module. The variables $\lambda_1$ and $\lambda_2$ were adjusted using a controlled variable method to find the relatively optimal results. The configuration of the neuron numbers in the smooth ODE follows the rule that the number of neurons in the second and third

layers equals the dimensionality of the environment actions, and the number of neurons in the first layer is greater than that of the latter two layers.

## F.1 TRAINING DETAILS ON VEHICLE TRAJECTORY TRACKING ENVIRONMENT

We employ the infinite-time approximate dynamic programming (INFADP) Li (2023), a model-based RL algorithm, for training in the vehicle trajectory tracking environment. We use the same hyperparameters for sine and double-line scenarios. The hyperparameters of INFADP are listed in Table 8.

Table 8: Algorithm hyperparameter

| Parameter | Setting |
|---|---|
| Replay buffer capacity | 1000000 |
| Buffer warm-up size | 1000 |
| Batch size | 64 |
| Discount $\gamma$ | 0.99 |
| Target network soft-update rate $\tau$ | 0.2 |
| Initial random interaction steps | 0 |
| Interaction steps per iteration | 8 |
| Network update times per iteration | 1 |
| Prediction step | 10 |
| Action bound | [-0.4, 0.4] |
| Exploration noise std. deviation | 0.2 |
| Hidden layers in input module | [64, 64] |
| Numbers of adaptive ODE neurons in each layer | [4, 2, 2] |
| Hidden layers in critic network | [64, 64] |
| Activations in critic network | ReLU |
| Optimizer | Adam |
| Actor learning rate | $1 \cdot 10^{-3}$ |
| Critic learning rate | $1 \cdot 10^{-3}$ |
| Weight $\lambda_1$ | $2 \cdot 10^{-2}$ |
| Weight $\lambda_2$ | $2 \cdot 10^{-3}$ |

## F.2 TRAINING DETAILS ON LQR

The classical optimal control problem is characterized as a linear quadratic regulation (LQR) problem with two-dimensional states and one-dimensional action. We use the INFADP algorithm to train this environment. The hyperparameters of INFADP are listed in Table 9.

## F.3 TRAINING DETAILS ON MUJOCO TASKS

Mujoco Todorov et al. (2012) is a simulation engine designed primarily for research in RL and robotics. It provides a versatile and physics-based platform for developing and testing various RL algorithms. Core features of Mujoco include a highly efficient physics engine, realistic modeling of dynamic systems, and support for complex articulated robots. Currently, it is one of the most recognized benchmark environments for RL and continuous control.

We use distributional soft actor-critic (DSAC)Duan et al. (2021), a model-free RL algorithm to train these eight robot control tasks. The hyperparameters of DSAC are listed in Table 10. The weights $\lambda_1, \lambda_2$ and numbers of smooth ODE neurons of each layer are listed in Table 11.

## G NETWORK MODULES AND TEST WEIGHT SELECTION METHODS

The output module is primarily used for complex control tasks. If the selected control task is relatively simple and has a small action space, such as the three-degree-of-freedom vehicle trajectory tracking control task in this paper, the output module is unnecessary. Additionally, the final test weights should

Table 9: Algorithm hyperparameter

| Parameter | Setting |
|---|---|
| Replay buffer capacity | 1000000 |
| Buffer warm-up size | 1000 |
| Batch size | 64 |
| Discount $\gamma$ | 0.99 |
| Target network soft-update rate $\tau$ | 0.2 |
| Initial random interaction steps | 0 |
| Interaction steps per iteration | 8 |
| Network update times per iteration | 1 |
| Prediction step | 1 |
| Action bound | [-5, 5] |
| Exploration noise std. deviation | 0.2 |
| Hidden layers in input module | [64, 64] |
| Numbers of adaptive ODE neurons in each layer | [2, 1, 1] |
| Hidden layers in critic network | [64, 64] |
| Activations in critic network | ReLU |
| Optimizer | Adam |
| Actor learning rate | $3 \cdot 10^{-5}$ |
| Critic learning rate | $8 \cdot 10^{-5}$ |
| Weight $\lambda_1$ | $2 \cdot 10^{-2}$ |
| Weight $\lambda_2$ | $2 \cdot 10^{-3}$ |

Table 10: Algorithm hyperparameter

| Parameter | Setting |
|---|---|
| Replay buffer capacity | 1000000 |
| Buffer warm-up size | 10000 |
| Batch size | 256 |
| Discount $\gamma$ | 0.99 |
| Initial alpha $\alpha$ | 0.27 |
| Target network soft-update rate $\tau$ | 0.005 |
| Initial random interaction steps | 0 |
| Interaction steps per iteration | 8 |
| Network update times per iteration | 1 |
| Prediction step | 1 |
| Action bound | [-1, 1] |
| Convolution kernel sizes (CarRacing) | [4, 3, 3, 3, 3, 3] |
| Convolution channels (CarRacing) | [8, 16, 32, 64, 128, 256] |
| Convolution strides (CarRacing) | [2, 2, 2, 2, 1, 1] |
| Convolution activation (CarRacing) | ReLU |
| Hidden layers in input module | [256, 256, 256] |
| Hidden layers in critic network | [256, 256, 256] |
| Activations in critic network | GeLU |
| Policy act distribution | TanhGauss |
| Policy min log std | -20 |
| Policy max log std | 0.5 |
| Policy delay update | 2 |
| Optimizer | Adam |
| Actor learning rate | $1 \cdot 10^{-4}$ |
| Critic learning rate | $1 \cdot 10^{-4}$ |
| Alpha learning rate | $3 \cdot 10^{-4}$ |
| Target entropy | - dim $(\mathcal{A})$ |

Table 11: Weight $\lambda_1, \lambda_2$ and numbers of smooth ODE neurons of each layer on Mujoco

| Env | weight $\lambda_1$ | weight $\lambda_2$ | Numbers of smooth ODE neurons |
|---|---|---|---|
| Humanoid-v3 | $1 \cdot 10^{-2}$ | $1 \cdot 10^{-2}$ | [20 17 17] |
| Pusher-v2 | $1 \cdot 10^{-3}$ | $1 \cdot 10^{-2}$ | [10 7 7] |
| Hopper-v3 | $1 \cdot 10^{-3}$ | $1 \cdot 10^{-2}$ | [6 3 3] |
| Reacher-v2 | $1 \cdot 10^{-2}$ | $1 \cdot 10^{-2}$ | [4 2 2] |
| Walker2d-v3 | $1 \cdot 10^{-2}$ | $1 \cdot 10^{-2}$ | [10 6 6] |
| Ant-v3 | $1 \cdot 10^{-5}$ | $1 \cdot 10^{-3}$ | [10 8 8] |
| InvertedDoublePendulum-v3 | $1 \cdot 10^{-2}$ | $1 \cdot 10^{-2}$ | [2 1 1] |
| CarRacing-v1 | $1 \cdot 10^{-3}$ | $1 \cdot 10^{-3}$ | [4 2 2] |

be selected based on a comprehensive evaluation of strategy performance and action smoothing inhibition ability, rather than simply adopting the training weights from the last iteration.

