# OpenReview forum: "ODE-based Smoothing Neural Network for Reinforcement Learning Tasks"
_ICLR.cc/2025/Conference — ICLR 2025 Spotlight_

### Official Review · Reviewer_DVZD · 2024-10-31

**Soundness:** 3
**Presentation:** 3
**Contribution:** 3
**Rating:** 8
**Confidence:** 3

**Summary:**

The authors propose an ODE that can serve as a neuron in a neural network. They then propose an actor-critic algorithm that incorporates the ODE neuron into the policy network, thereby allowing for smoother action outputs compared to existing methods. The authors present relevant theoretical results as well as extensive empirical validation that showcases the usefulness of their method.

**Strengths:**

This paper provides a theoretically-sound method for smoothing the action output, which, based on the empirical results, leads to better performance overall. The authors provide useful and sound theoretical results, as well as convincing empirical evidence that shows how their method generally outperforms existing methods. The paper is polished, and presented in an easy-to-read manner. Overall, this paper provides a solid contribution.

**Weaknesses:**

The authors heavily rely on the Action Fluctuation Ratio (AFR) as the key metric for their implementation and analysis. Given that the AFR was only proposed recently (in Song et al., 2023), the paper would benefit from a more thorough discussion on why this metric is an acceptable one to use. In fact, one could argue that if there is no universally-agreed upon metric, perhaps the authors should consider multiple candidates for metrics in their analysis and justify (in the paper) why AFR is the best one to use.

Similarly, aside from MPC, the other baselines used in the experiments are not well-motivated in the text. In particular, it should be made clearer why the baselines used are the correct ones to use, and why they constitute a reasonably complete set of relevant baselines to consider.

Finally, in lines 231-232, “We also think that…” should be framed in a more scientific manner. Appendix C is not convincing either (while it shows an evidence).

**Questions:**

1.	Theorem 2 is based on the specific setup in Eq (12) according to line 245 on p.5. But the proof does seem to rely on Eq (12). Can you discuss the generalizability of your main results.
2.	How do the terms added to the loss in Equation 16 affect the performance? What are the guidelines on how to set the two weights (\lambda_1 and \lambda_2)?
3.	What is the computational burden due to the ODE solver?

---

> ### Author Response · Authors · 2024-11-21
> **[1/3] Rebuttal by Authors**
>
> We thank you for the careful reading of our paper and constructive comments in detail.
>
> ### **> Weakness 1**
> It may be that my inadequate quoting has caused you concern. In fact, Mysore et al. (2020) [1] were **the first to introduce the use of $\Delta a_t = a_t - a_{t-1}$ to define the degree of action smoothing**, which became a key performance metric. The exact formulation is provided in Section 5, "Evaluation," of their paper. Additionally, Chen et al. (2021) [2] employed the AFR as a central measure of action smoothing. A detailed description is available in Section 4.1, "Training and Evaluation," of their paper. However, their work focuses on AFR as a metric for discrete action spaces, while LipsNet (2023) [3] extends it to continuous action spaces. There are also some works [4-6] that directly or indirectly add AFR as a penalty term in the training loss function or reward function to improve the smoothness of the control action, which shows that **AFR is a very important index for measuring the smoothness of the action.**
>
> After giving numerous citations, we hope you will recognize that this is a generally agreed upon metric, and is one of **the most intuitive and effective performance indicators of control smoothness in practical control tasks.**
>
> ---
>
> But now that you mention it, we found another test metric named mean weighted frequency (MWF) in [4] [7].
>
> When analyzed in the frequency domain, the unsmoothness of the control policy can be explained by the high-frequency components in the action sequence. Given an action sequence $\{a_{0},a_{1},\cdots a_{T}\}\sim\rho_{\pi}$, the MWF is defined
>
> $\zeta(\pi)=\mathbb{E} _ {\{a_0,a_1,\cdots a_T\}\sim\rho_\pi}\left[\sum _ {i=1}^n\frac{\sum _ {j=1}^kA_{ij}f_{ij}}{n\sum _ {j=1}^kA_{ij}}\right],$
>
> where $A_{ij}$ and $f_{ij}$ is the $j$-th amplitude and frequency component in the frequency spectrum of the $i$-th dimension action sequence.
>
>
> We selected four robot control tasks, Humanoid-v3, Walker2d-v3, Pusher-v2 and Hopper-v3, and used the MWF indicator for experimental testing. Since the state values of different Mujoco tasks vary greatly, we set two levels of uniform noise for the four tasks, as shown in Table1. For the whole tasks, noise is added to all states.
>
> **Table 1 Different levels of uniform noise for different Mujoco tasks.** The numbers in the table represent the range of values for uniform noise.
> | Noise level | Humanoid-v3   | Walker2d-v3   | Pusher-v2     | Hopper-v3     |
> | ----------- | ------------- | ------------- | ------------- | ------------- |
> | level 1     | [-0.05, 0.05] | [-0.15, 0.15] | [-0.10, 0.10] | [-0.05, 0.05] |
> | level 2     | [-0.10, 0.10] | [-0.20, 0.20] | [-0.20, 0.20] | [-0.07, 0.07] |
>
> The comparison network structures used in the experiment are MLP, MLP-LowFilter and SmODE, where **LowFilter is a low-pass filter that utilizes a three-step history state.** The results, which are the averages of five seeds, are shown in Table 2.
>
> **Table 2 Experimental results.** Average control performance of MLP, MLP-LowFilter, and SmODE for different uniform noise levels, where level 1 is on the left column and level 2 is on the right column. The average mean weighted frequency is indicated in parentheses. Results are expressed as mean ± standard deviation of five independent environmental seeds.
> | Network structure | Humanoid-v3                             | Walker2d-v3                               | Pusher-v2                          | Hopper-v3                                  |
> | ----------------- | --------------------------------------- | ----------------------------------------- | ---------------------------------- | ------------------------------------------ |
> | MLP               | 10004±77 (10.66Hz)  9968±45 (11.69Hz)   | 4959±620 (26.12Hz) 964±705 (28.13Hz)      | -76±9 (4.82Hz)  -96±6 (5.14Hz)     | 1669±430 (24.35Hz)  1425±346 (26.17Hz)     |
> | MLP+LowFilter     | 9428±79 (9.32Hz)  9559±106 (10.12Hz)    | 4444±679 (22.76Hz) 4066±877 (24.06Hz)     | -44±2 (4.06Hz)  -81±6 (4.58Hz)     | 1338±542 (22.39Hz)  1185±582 (22.67Hz)     |
> | SmODE             | **10552±36 (7.45Hz) 10478±18 (8.22Hz)** | **5678±607 (20.05Hz) 5585±705 (22.37Hz)** | **-37±4 (3.31Hz)  -42±3 (4.03Hz)** | **3331±492 (23.32Hz)  3249±611 (23.71Hz)** |
>
> The experimental results demonstrate that **AFR and MWF provide consistent measures of action smoothness, with both serving as effective indicators.** AFR, being more intuitive and easier to calculate, has been widely used in related works since 2020. Consequently, we selected AFR as the primary indicator for assessing action smoothness.

---

> ### Author Response · Authors · 2024-11-21
> **[2/3] Rebuttal by Authors**
>
> ### **> Weakness 2**
> Thank you for your suggestion. Next, we will explain why we choose MLP, LipsNet, and LTC as the baseline network structure and why they can form complete related baselines.
>
> First, let's discuss MLP. In deep reinforcement learning (DRL) control algorithms, the most commonly used policy network structure is the MLP. **This structure also represents the core issue we aim to address: the problem of action oscillation that arises when using classic DRL algorithms for control.** Thus, using MLP as the baseline network is essential for evaluating the performance and smoothness improvements achieved by alternative network structures.
>
> The second one is LipsNet, **which is the SOTA algorithm of smooth neural network.** Our paper uses it as the core comparison algorithm. If we can achieve better performance and smoothness than LipsNet in most control tasks, it can strongly prove that our method is effective and has more advantages.
>
> Finally, LTC is one of the classic algorithms of Neural ODE, and it is mentioned in the paper that it has a certain anti-interference ability. Therefore, it is also very important to **compare with LTC and achieve better performance and smoothness, which belongs to the same category of Neural ODE.** In addition, it can further show that we are the first to solve the two causes of the unsmoothness of DRL control actions (high-frequency noise interference and unconstrained Lipshchitz constant) at the same time. I believe this is very interesting for the RL community.
>
> ### **> Weakness 3**
> Your suggestion is very good, we have deleted this unscientific statement. From the results of the ablation experiment, we can see that the adaptive constraint on the Lipschitz constant of the neural network plays a great role in the smoothness of the control action and does not affect the performance of the strategy. This is due to the fact that $g(x(t), I(t), t, \theta)$ can adjust the degree of Lipschitz constraint according to the current input and state, as described in Theorem 2. The image in Appendix C is intended to demonstrate the ability to adaptively adjust the degree of Lipschitz constraint based on the speed of dynamic changes in the current state.
>
> ### **> Question 1**
> Your discovery is very sharp. In the proof of Theorem 2, we did use a specific parameter range, but in fact our proof does not depend on the specific parameter selection. As long as $f(x(t), I(t), t, \theta)>0$ and is bounded, the proof of Theorem 2 is valid.
>
> ### **> Question 2**
> Your suggestion is excellent. In the Walker2d-v3 task, we replaced the policy network in the TD3 algorithm with SmODE and conducted sensitivity experiments on the hyperparameters $\lambda_1$ (Filtering Hyperparameter) and $\lambda_2$ (Lipschitz Hyperparameter).
>
> Under this setup, we tuned the hyperparameters and found that the optimal values were both 0.01. To explore the sensitivity within the same order of magnitude, we experimented with 0.02; for different orders of magnitude, we experimented with 0.001 and 0.1. When analyzing the sensitivity of one parameter, the other parameter is kept constant at 0.01. Gaussian noisy variance is 0.1. The results are shown in Table 3:
>
>
> **Table 3 Hyperparameter sensitivity analysis.** When analyzing the sensitivity of one parameter, the other parameter is kept constant at 0.01. Gaussian noisy variance is 0.1.
> | **Hyperparameter** | **Value** | **Performance**     |
> | ------------------ | --------- | ------------------- |
> | $\lambda_1$        | 0.01      | **3504±773 (1.11)** |
> |                    | 0.1       | 2471±576 (0.87)     |
> |                    | 0.001     | 3609±461 (1.17)     |
> |                    | 0.02      | 3410±671 (1.08)     |
> | $\lambda_2$        | 0.01      | **3504±773 (1.11)** |
> |                    | 0.1       | 2154±615 (0.64)     |
> |                    | 0.001     | 3720±428 (1.46)     |
> |                    | 0.02      | 3280±561 (0.99)     |
>
> The results indicate that $\lambda_2$ has a stronger impact on action smoothness compared to $\lambda_1$. Within the same order of magnitude, the differences in performance and action smoothness are not significant, but across different orders of magnitude, there are substantial differences.
>
> ---
>
> The selection of parameters is divided into two steps:
> 1. First, set $\lambda_1$ to 0, and select the weight of $\lambda_2$ to 0.01 and 0.001 for experiments to determine which weight can better balance strategy performance and smoothness.
> 2. Fix $\lambda_2$ unchanged, and change the value of $\lambda_1$ from the same as $\lambda_2$ to 10 times smaller for experiments to determine which weight can better balance strategy performance and smoothness.
>
> Tip: The weight of $\lambda_2$ is greater than or equal to the weight of $\lambda_1$.

---

> ### Author Response · Authors · 2024-11-21
> **[3/3] Rebuttal by Authors**
>
> ### **> Question 3**
> Your suggestion is great. We provide **an additional test on the backward and forward time.** The platform used is an AMD Ryzen Threadripper 3960X 24-Core Processor. The network structures compared are MLP, SmODE, MLP-SN [8] and LipsNet [3]. The latter two algorithms are smoothing neural networks. The power iteration step in MLP-SN is set to 15. The experimental results are shown in Table 4.
>
> **Table4 Forward and Backward Propagation Time Test.** The experimental comparison algorithms are MLP, SmODE, MLP-SN and LipsNet.
> | **Mode** | **Batch size** | **MLP** | **SmODE** | **MLP-SN** | **LipsNet** |
> | -------- | -------------- | ------- | -------- | ---------- | ----------- |
> | Forward  | 1              | 0.10 ms | 0.25 ms  | 0.11 ms    | 0.75 ms     |
> |          | 100            | 0.11 ms | 1.17 ms  | 0.12 ms    | 1.41 ms     |
> | Backward | 1              | 0.17 ms | 0.33 ms   | 2.98 ms    | 0.45 ms     |
> |          | 100            | 0.28 ms | 0.64 ms   | 3.08 ms    | 0.73 ms     |
>
> It implies that the computation time of SmODE is high than MLP's. Fortunately, the 1-batch forward time is a small value within 1 ms, which still **acceptable in the real-time application.** We found that the training time of SmODE is about 2-3 times longer than that of MLP. The bottleneck is the backpropagation through time. It is actually **what we are working on recently.** We are going to introduce an **Adjoint method** to eliminate the bottleneck. We would advise to follow our future work.
> ### **> Reference**
> [1] Mysore et al. "How to Train your Quadrotor: A Framework for Consistently Smooth and Responsive Flight Control via Reinforcement Learning", ACM Transactions on Cyber-Physical Systems, 2020.
>
> [2] Chen et al. "Addressing Action Oscillations through Learning Policy Inertia", AAAI, 2021.
>
> [3] Song et al. "LipsNet: A Smooth and Robust Neural Network with Adaptive  Lipschitz Constant for High Accuracy Optimal Control", ICML, 2023.
>
> [4] Mysore et al. "Regularizing Action Policies for Smooth Control  with Reinforcement Learning", IEEE ICRA, 2021.
>
> [5] Shen et al. "Deep Reinforcement Learning with Robust and Smooth Policy", ICML, 2020.
>
> [6] Yu et al. "TAAC: Temporally Abstract Actor-Critic for Continuous Control", NeurIPS, 2021.
>
> [7] Wang et al. "Smooth Filtering Neural Network for  Reinforcement Learning", IEEE TIV, 2024.
>
> [8] Miyato et al. "Spectral normalization for generative adversarial networks," arXiv, 2018.

---

### Official Review · Reviewer_7dLr · 2024-11-03

**Soundness:** 3
**Presentation:** 2
**Contribution:** 3
**Rating:** 5
**Confidence:** 3

**Summary:**

This paper presents a Smooth Ordinary Differential Equation-based Neural Network (SmODE) architecture, designed to address the issue of unsmooth control actions in deep reinforcement learning (RL). The SmODE network mitigates high-frequency input noise and restricts the neural network’s inherent high Lipschitz constants by employing a dual mechanism that combines low-pass filtering with Lipschitz constant control. This approach enhances the smoothness and robustness of the policy.

**Strengths:**

1. **Innovative Approach**
    - **First Combination of Neural ODE to Address Dual Issues**:
        - This work is the first to attempt using Neural Ordinary Differential Equations (Neural ODE) to simultaneously address high-frequency input noise and action unsmoothness caused by the network’s Lipschitz constants. The SmODE network provides a comprehensive solution by integrating low-pass filtering with Lipschitz constant control, effectively improving policy smoothness and robustness rather than optimizing for a single issue alone. The design cleverly mimics the structure of a classical first-order low-pass filter but replaces the fixed time constant with a state-dependent learnable function. This not only retains the noise suppression capability of the low-pass filter but also enhances the model's adaptability and dynamic response through the learning mechanism. Additionally, it leverages the continuity properties of Neural ODEs.
    - **Dual Smoothing Mechanism**:
        - The SmODE network introduces learnable state-dependent time constants \(\tau(x)\) and state mapping functions \(g(x(t), I(t), t, \theta)\), controlling the rate of state change via the time constants and regulating the Lipschitz constants through the state mapping functions. This simultaneously suppresses high-frequency noise and controls the network’s sensitivity, achieving smooth and robust action outputs.
    - **Biologically-Inspired Neuron Design**:
        - SmODE neurons emulate the characteristics of biological neurons by combining synaptic weights (\(w_{ij}\)), membrane capacitance (\(C_{mi}\)), and resting potential (\(x_{leaki}\)). Through low-pass filtering, they enable adaptive adjustment of state boundaries. This design not only enhances the biological plausibility of the model but also improves its performance and stability in practical control tasks.
2. **Technical Depth**
    - **Theoretical Foundation**:
        - The paper provides formal theorems and proofs (Theorem 1 and Theorem 2), theoretically demonstrating the bounds on hidden states and the upper limits of derivatives, showcasing a deep understanding of Neural ODEs and control theory.
    - **Extensive Experimental Tasks**:
        - The authors conducted comprehensive experimental validations across various reinforcement learning tasks, including vehicle trajectory tracking, linear-quadratic regulation problems, and eight robot control tasks in Mujoco. The experimental results demonstrate that the SmODE network significantly outperforms traditional Multi-Layer Perceptrons (MLP) and methods like LipsNet in terms of action smoothness and noise robustness, showcasing superior control performance.
    - **Validation of Key Component Contributions**:
        - Ablation studies demonstrate the significant contributions of the time constant term and state boundary adjustment term in enhancing action smoothness. By individually removing these key components, the experimental results show a substantial degradation in the smoothness effect of SmODE, further validating the rationality and effectiveness of its design choices.

**Weaknesses:**

1. **Insufficient Description of the Core Neuron Model (Equation 12)**:
    - While similar works are referenced in the paper, the transition from the general smooth ODE neuron model (Equation 10) to the specific biological neuron model (Equation 12) deserves more detailed exposition, as it serves as the core design of the SmODE network. The roles, value ranges, and impacts on model performance of key parameters (\(w_{ij}\), \(C_{mi}\), \(\gamma_{ij}\), \(\mu_{ij}\)) are not thoroughly discussed. A more comprehensive explanation would enhance the paper's logical flow and motivation. Moreover, the introduction of multiple parameters in Equation 12 increases the model's complexity and raises concerns about the reliability of experimental results due to the extensive parameter tuning space. The introduction of multiple parameters in Equation 12 increases the model's complexity and the difficulty of parameter tuning. There is a lack of clear guidelines or empirical rules for selecting these parameters.
2. **Incomplete Theoretical Assumptions and Proofs**:
    - **Missing Boundedness Assumption for Function \( g(\cdot) \)**:
        - Theorem 1 asserts that the hidden state of the neuron is bounded by the maximum and minimum values of the function \( g(x(t), I(t), t, \theta) \), but it does not sufficiently explain or prove whether \( g(\cdot) \) itself is bounded. If \( g(\cdot) \) is unbounded, the validity of the theorem is questionable.
    - **Incompleteness and Rigorousness of Theorem Proofs**:
        - **Theorem 1**: The proof lacks a detailed discussion on the boundedness of \( g(\cdot) \) and does not adequately explain why \( \frac{dx_i}{dt} \leq 0 \) holds in all cases. The derivation steps for the lower bound \( \min(0, g(\cdot)_{\min_i}) \leq x_i(t) \) are not sufficiently clear.
        - **Theorem 2**: The derivation process lacks detailed explanations regarding the boundary control of \( f(x(t), I(t), t, \theta) \) and \( g(x(t), I(t), t, \theta) \). The origin of the constant \( C \) and its relationship with \( M(\cdot)_i \) are not clearly explained, resulting in the final inequality \( \left| \frac{dx_i(t)}{dt} \right| \leq M(\cdot)_i \cdot C \) lacking sufficient mathematical justification.
3. **Insufficient Discussion on Computational Efficiency**:
    - Although the paper mentions that the training time of SmODE increases due to the use of numerical ODE solvers, it lacks an in-depth discussion of this computational overhead and an analysis of its impact on scalability and real-time performance.
4. **Lack of Direct Comparisons with Some Related Works**:
Related research focuses on addressing action smoothness in reinforcement learning, with methodologies highly relevant to SmODE. Without comparing SmODE to these methods, readers would find it difficult to understand SmODE's specific advantages.
    - Related similar studies such as **"Smooth Filtering Neural Network for Reinforcement Learning"** are not included in the comparisons, potentially missing opportunities to demonstrate the relative advantages of SmODE against the latest methods.
    - Recent relevant methods like **Neural CDEs (2020)** and **Stable Neural Flows (2021)** are not compared experimentally. These methods are relevant to handling continuous-time sequences and controlling stability, aligning closely with the goals of SmODE.

**Questions:**

1. **Provide a Detailed Description of Equation 12**:
    - Offer a comprehensive explanation of each parameter in Equation 12 (\(w_{ij}\), \(C_{mi}\), \(\gamma_{ij}\), \(\mu_{ij}\)), including their the influences of value ranges, and specific roles in controlling action smoothness. Provide additional theoretical support or empirical results to demonstrate how the introduction of these parameters influences the model's performance.
2. **Discuss Computational Efficiency and Scalability**:
    - Provide a more detailed analysis of the computational overhead, the training time and inference speed of the SmODE network. Explore the influences of the ODE solving process or the number of iterations, to improve the model’s computational efficiency and real-time performance.
3. **Expand Experimental Comparisons and Discussions**:
    - [**Smooth Filtering Neural Network for Reinforcement Learning**](https://ieeexplore.ieee.org/abstract/document/10643291/): This work is highly relevant to the current paper and should be prominently compared to highlight the advantages and characteristics of the proposed method.

---

> ### Author Response · Authors · 2024-11-20
> **[1/3] Rebuttal by Authors**
>
> We appreciate the reviewer for the careful reading of our paper and detailed discussions.
>
> ### **> Weakness 1 & Question 1**
> As you can sense, “Bionic modeling” can be confusing at first glance. We elaborate on the cited article: Neural circuit policies (NCP) enabling auditable autonomy [1]. The researchers in this work **reference the structure of the nervous system of *Caenorhabditis elegans*,** a nematode renowned for its near-optimal neural network architecture and coordinated neural information processing mechanisms. NCP mentions that bionic modeling of neurons using imitation *Caenorhabditis elegans* can enhance their computational capabilities. Therefore, **biological concepts such as membrane capacitance are mentioned in our paper as learnable parameters with an initial range of values consistent with those in the NCP.**
>
> ---
>
> Since you are very concerned about the impact of the value range of biological modeling parameters on performance, we conducted sensitivity experiments on the four parameters $w_{ij}, C_{m_{i}}, \mu_{ij}, \gamma_{ij}$ in the Pusher-v2 robot control task. With all other parameters held constant, two sets of experiments **were conducted by expanding and shrinking the value range of each parameter by a factor of 10, resulting in a total of 8 experiments.** The experiment used two different levels of uniform noise, [-0.1, 0.1] and [-0.2, 0.2], and the results are shown in Table 1.
>
>
> **Table1 Experimental results.**. The average action fluctuation rate is indicated in parentheses. Results are expressed as mean ± standard deviation of five independent environmental seeds.
> | Parameters/Noise level     | [-0.1, 0.1]  | [-0.2, 0.2]  |
> | -------------------------- | ------------ | ------------ |
> | MLP                        | -76±9 (1.36) | -96±6 (1.57) |
> | Standard SmODE             | -37±4 (0.39) | -42±3 (0.89) |
> | $w_{ij}\in(0.0001, 0.1)$   | -45±3 (0.49) | -50±4 (1.09) |
> | $w_{ij}\in(0.01, 10)$      | -41±3 (0.28) | -49±5 (0.72) |
> | $C_{m_{i}}\in(0.04, 0.06)$ | -31±2 (0.41) | -33±3 (0.93) |
> | $C_{m_{i}}\in(4, 6)$       | -46±3 (0.31) | -49±2 (0.78) |
> | $\gamma_{ij}\in(0.3, 0.8)$ | -42±5 (0.30) | -45±6 (0.93) |
> | $\gamma_{ij}\in(30, 80)$   | -33±6 (0.38) | -51±3 (0.66) |
> | $\mu_{ij}\in(0.03, 0.08)$  | -38±2 (0.36) | -46±2 (0.91) |
> | $\mu_{ij}\in(3, 8)$        | -50±5 (0.14) | -55±1 (0.29) |
>
> The experimental results show that adjusting the value range of biological parameters will affect both the performance and smoothness of the strategy, resulting in a classic seesaw effect. **However, the performance and smoothness of all experiments exceeded that of MLP.**
>
> ---
>
> Equation 12 introduces several parameters that, while increasing the model's complexity, **enhance its biological plausibility and strengthen the connection between Neural ODEs and biological structures.** Despite this added complexity, **no manual hyperparameter tuning was performed for the biological modeling parameters** in any of the experiments presented in this paper. Nevertheless, the experimental results consistently demonstrated high performance and effective smoothing capabilities. Consequently, there is no extensive parameter tuning space that would undermine the reliability of the results. Although the absence of clear empirical rules may seem concerning, it is important to note that none of our experiments required adjustments to the biological modeling parameters. Furthermore, **their range remained consistent with that of NCP**, which is why we did not explore empirical tuning rules.
>
> ### **> Weakness 2**
> **2.1**
>
> You can look at line 242. We define $g(x(t), I(t), t, \theta)=\text{tanh}(h(x(t), I(t), t, \theta))$, where tanh stands for a hyperbolic tangent function, and $h(x(t), I(t), t, \theta)$ stands for a neural network. **Thus, $g(x(t), I(t), t, \theta)$ takes values in the range (-1, 1).** Therefore there is no problem of lack of boundedness assumption.
>
> **2.2**
>
> *2.2.1*
>
> Our definition of $g(x(t), I(t), t, \theta)$ already **guarantees its boundedness.**
>
> In addition, you may have misunderstood. In the proof, we **only examine the sign of $\frac{dx_i}{dt}$ at the special point $x_i(t) = M = \max(0,g(\cdot)_i^{\max})$, instead of requiring $\frac{dx_i}{dt} \leq 0$ for all values ​​of $x_i(t)$.** In order to prove that the upper bound $x_i(t) \leq \max(0,g(\cdot)_i^{\max})$ holds, we only need to prove that: when $x_i(t)$ reaches this upper bound, its derivative is non-positive, which ensures that the trajectory will not exceed this upper bound. In fact, the state of the neuron can change freely between the lower and upper bounds, and **the derivative can be positive or negative.**
>
> The derivation steps for the lower bound are completely symmetrical. **You only need to replace $M = \max(0,g(\cdot)_i^{\max})$ with $M = \min(0,g(\cdot)_i^{\min})$.** At this time, the derivative is greater than 0, and the proof is complete.

---

> ### Author Response · Authors · 2024-11-20
> **[2/3] Rebuttal by Authors**
>
> *2.2.2*
>
> Our definition of $g(x(t), I(t), t, \theta)$ already **guarantees its boundedness.** In addition, you can see that $f(x(t),I(t),t,\theta) = \frac{w_{ij}}{C_{m_{i}}} \text{sigmoid}(\cdot)$ is defined in line752, where $w_{ij}\in(0.001,1.0), C_{m_{i}}\in(0.4, 0.6), sigmoid(\cdot)\in(0, 1)$. Therefore, we can get $0 < f(x(t),I(t),t,\theta) < \frac{1.0}{0.4} = 2.5$.
>
> In fact, we can clearly point out that C = 5, which is determined by the upper bound of f: $C = 2 \cdot \max(f(x(t),I(t),t,\theta)) = 2 \cdot 2.5 = 5$. **Therefore, $M(x(t), I(t), t, \theta)$ has no direct connection with $C$.** I believe that after giving the above detailed explanation, you can believe that **our work has a solid theoretical foundation.**
>
> ### **> Weakness 3**
> You are right that the discussion of scalability, computational overhead, and real-time performance is very important.
>
> Regarding scalability, I first introduce the design principles of the number of neurons in the three layers of Smooth ODE Module. You can see Page 19 of the paper, Table 11. The number of neurons in **the second and third layers is the same as the action dimension of the control task**, while the number of neurons in **the first layer is slightly larger than the action dimension**. The action dimension of the actual control task is not very large, so **the number of neurons in the Smooth ODE Module will not be very high.**
>
> To address the issues of training efficiency and inference efficiency, we provide **an additional test on the backward and forward time.** The platform used is an AMD Ryzen Threadripper 3960X 24-Core Processor. The network structures compared are MLP, SmODE, MLP-SN [2] and LipsNet [3]. The latter two algorithms are smoothing neural networks. The power iteration step in MLP-SN is set to 15. The experimental results are shown in Table 2.
>
> **Table2 Forward and Backward Propagation Time Test.** The experimental comparison algorithms are MLP, SmODE, MLP-SN and LipsNet.
> | **Mode** | **Batch size** | **MLP** | **SmODE** | **MLP-SN** | **LipsNet** |
> | -------- | -------------- | ------- | -------- | ---------- | ----------- |
> | Forward  | 1              | 0.10 ms | 0.25 ms  | 0.11 ms    | 0.75 ms     |
> |          | 100            | 0.11 ms | 1.17 ms  | 0.12 ms    | 1.41 ms     |
> | Backward | 1              | 0.17 ms | 0.33 ms   | 2.98 ms    | 0.45 ms     |
> |          | 100            | 0.28 ms | 0.64 ms   | 3.08 ms    | 0.73 ms     |
>
> It implies that the computation time of SmODE is high than MLP's. Fortunately, the 1-batch forward time is a small value within 1 ms, which still **acceptable in the real-time application.** We found that the training time of SmODE is about 2-3 times longer than that of MLP. The bottleneck is the backpropagation through time. It is actually **what we are working on recently.** We are going to introduce an **Adjoint method** to eliminate the bottleneck. We would advise to follow our future work.
>
> ### **> Weakness 4 & Question 3**
> **4.1**
>
> Thank you for doing your extensive reading and finding this similar study that was done very close in time. **I need to remind you that according to the last QA of [ReviewerGuide](https://iclr.cc/Conferences/2025/ReviewerGuide), this work published on 2024.8.21 (after 2024.7.1) is a concurrent paper and does not need to be compared.** However, we found the work of Wang et al. [6] to be quite interesting. After reading it, we will compare and analyze it from theoretical perspective, discuss its advantages and disadvantages.
>
> Theoretically, their work addresses the issue of unsmooth action in neural network control solely from the perspective of filtering, **while overlooking the fact that the unconstrained Lipschitz constant of the neural network itself is also one of the core reasons for the unsmooth action.** This work can be considered to use the simplest Neural ODE, using a first-order forward Euler expansion. Our Neural ODE uses the bionic modeling approach, has stronger representation capabilities, and uses a semi-implicit Euler discretization with higher solution accuracy.
>
> In terms of experimental settings, they **conducted training in a noisy environment by artificially adding noise**, which is quite different from the basic RL training environment. In addition, they need to **retain the hidden state of 4 steps of history for filtering**, which makes it difficult for their algorithm to be seamlessly integrated into other RL algorithms and frameworks. In summary, they gave some additional designs compared to SmODE in order to adapt to Gaussian noise interference.

---

> ### Author Response · Authors · 2024-11-20
> **[3/3] Rebuttal by Authors**
>
> **4.2**
>
> *4.2.1*
>
> Neural CDEs [4] address a fundamentally different problem from ours. **While we focus on ensuring smooth control actions in reinforcement learning by designing a neural network with ODEs to reduce oscillations, their work tackles the issue of irregular sampling in time series data, using CDEs to manage time dimension irregularities.**
>
> In theory, our paper builds on Neural ODEs, with the key innovation being the use of low-pass filtering and Lipschitz constraints to achieve smoothness. In contrast, their work is based on CDE theory, with the primary innovation being the handling of time dependencies in sequences through control differential equations.
>
> Regarding application, our focus is on control problems in **reinforcement learning**, such as robot control and autonomous driving, while their work targets **general time series modeling tasks**, including sensor and medical data.
>
> In summary, this method is not suitable for our experimental comparison. (You may have noticed that in Section 3.4 of their work, they proposed ***Training via the adjoint method*** to reduce memory overhead and speed up training. This is exactly the method we want to use in our subsequent work. Thank you for your suggestion.)
>
> *4.2.2*
>
> We appreciate your suggestion to compare our work with Stable Neural Flows (SNF) [5]. **While SNF offers valuable theoretical insights into designing stable Neural ODEs, its focus is on providing stability guarantees through energy functions.**
>
> Our work and SNF are complementary: SNF provides a theoretical foundation for stable Neural ODEs, whereas our approach delivers a concrete implementation and experimental validation of smooth control using an ODE-based neural architecture. **The absence of implementation code in SNF makes direct experimental comparison infeasible.** However, we believe that integrating SNF's theoretical framework with our practical implementation could be a promising direction for future work, combining theoretical stability guarantees with smooth control capabilities. In summary, this method is not suitable for our experimental comparison.
>
>
>
> ### **> Question 2**
> In Weakness3's response, we thoroughly analyzed and compared the computational and reasoning times of SmODE with those of other network structures. We demonstrated that SmODE's 1-batch reasoning time is under 1 ms, which satisfies practical control requirements. Next, we analyze the impact of the number of ODE solver iterations on strategy performance and smoothness.
>
> We experimented with three settings for the Pusher-v2 task, 3, 6 (Standard SmODE), and 9 iterations. We set two levels of uniform noise for the Pusher-v2 task. They are [-0.1, 0.1], [-0.2, 0.2]. The experimental results are shown in Table 3.
>
> **Table3 Experimental results.** The average action fluctuation rate is indicated in parentheses. Results are expressed as mean ± standard deviation of five independent environmental seeds.
> | Solver iteration/Noise level | [-0.1, 0.1]  | [-0.2, 0.2]  |
> | ---------------------- | ------------ | ------------ |
> | MLP                        | -76±9 (1.36) | -96±6 (1.57) |
> | SmODE-3                | -38±4 (0.72) | -53±6 (1.13) |
> | SmODE-6                | -37±4 (0.39) | -42±3 (0.89) |
> | SmODE-9                | -33±6 (0.31) | -43±4 (0.81) |
>
> Experimental results show that increasing the number of iterations can indeed improve the accuracy of the solution, thereby improving the performance and smoothness of the strategy, but it also faces the risk of longer inference and training time. Compared with the 3-step iteration to the 6-step iteration, **the smoothness improvement from 6 to 9 is relatively small**. In addition, compared with MLP, the 6-step iteration has achieved a great improvement in smoothness. In summary, the 6-step iteration solution is a good choice.
>
> ### **> Reference**
> [1] Lechner et al. "Neural circuit policies enabling auditable autonomy", Nature machine intelligence, 2020.
>
> [2] Miyato et al. "Spectral normalization for generative adversarial networks," arXiv, 2018.
>
> [3] Song et al. "LipsNet: A Smooth and Robust Neural Network with Adaptive  Lipschitz Constant for High Accuracy Optimal Control", ICML, 2023.
>
> [4] Kidger et al. "Neural Controlled Differential Equations for Irregular Time Series", NeurIPS, 2020.
>
> [5] Massaroli et al. "Stable Neural Flows", arXiv, 2020.
>
> [6] Wang et al. "Smooth Filtering Neural Network for  Reinforcement Learning", IEEE TIV, 2024.

---

> ### Comment · Reviewer_7dLr · 2024-11-20
> **Potential Academic Integrity Concerns**
>
> I would like to raise concerns regarding the publication timeline and technical overlap of submission #7391. While the IEEE paper's publication date (Aug 21, 2024) comes after the submission deadline and falls within the concurrent work window, there are serious concerns about the substantial technical overlap between these papers that warrant further investigation by the program committee.
>
> The fundamental technical framework of both papers shows remarkable similarities. Both papers propose essentially identical solutions to the smoothing RL control actions problem. Although different mathematical notation is used, the underlying principles remain the same. Notably, the ODE-based implementation effectively realizes the same first-order low-pass filtering mechanism as in the IEEE paper.
>
> The methodological approach also demonstrates significant overlap. Both works employ learnable state-based parameters for filtering and focus on dynamic filtering of hidden states. The treatment of high-frequency noise follows similar principles, and both papers ultimately achieve comparable results through analogous technical means.
>
> The authors' rebuttal statement that the published work "addresses the issue solely from the perspective of filtering" appears to understate these similarities. Both papers utilize equivalent mathematical principles, demonstrate similar experimental results, and share core technical approaches. The distinction between ODE-based and filtering-based approaches becomes particularly thin when examining the actual implementation details.
>
> Given these substantial overlaps, I recommend that the program committee investigate this submission for potential academic integrity concerns. A proper investigation would help ensure the integrity of the peer review process and maintain ICLR's high standards.

---

> ### Author Response · Authors · 2024-11-25
>
> Dear reviewer 7dLr,
>
> It is less than 2 days before the end of the discussion phase. I sincerely hope that you can take some time out of your busy schedule to check some of our responses. **Your questions have given us a lot of thought**, and we look forward to having in-depth exchanges with you or reaching an agreement on the contribution of this work.

---

> ### Author Response · Authors · 2024-12-03
>
> Through careful deliberation and analysis of the distinctions in theoretical frameworks and experimental protocols, we present **objective evidence that carries greater scholarly weight than subjective comparative evaluations.** Next, we shall demonstrate through the following rigorous arguments that our work is entirely free of academic integrity concerns. (Incidentally, you did not make any valid replies during the three-week discussion period, and on the last day of the discussion phase you revised the visible scope of the review via Revisions, and we did not receive any notifications. **Your evident expertise in manipulating OpenReview's notification protocols astounds us.**)
>
> 1. Although both are first-order low-pass filtering, the work you mentioned is a **discrete-time first-order low-pass system in fixed form**, while the Neural ODE proposed in this paper is a **continuous-time first-order low-pass system in variable form**. Therefore, the latter has a higher degree of freedom in neuron structure design and can embed more good theoretical properties (e.g., adaptive Lipschitz constant constraints, adaptive filtering degree adjustment proposed in this paper) according to the demand.
>
> 2. The two algorithms are also very different in terms of training approach, as the work you mentioned uses the most basic backpropagation, whereas our work solves continuous-time systems using semi-implicit Eulerian discretization, thus requiring the use of the **BPTT method to propagate the gradient.**
>
> 3. As you mentioned in ***Strength***, our work are the first combination of Neural ODE to address dual issues leading to unsmooth control, and the work you mentioned does not consider the neural network Lipschitz constant unsmooth control problem. This is because they use fixed form discrete first-order low-pass systems that **do not have the ability to design adaptive Lipschitz constant constraints.**
>
> 4. The work you mention uses hundreds or thousands of filter neurons, whereas our work requires only a dozen or so Smooth ODE units.
>
> 5. The core mathematical formula for the work you mention is $\tau\frac{\mathrm{d}h(t)}{\mathrm{d}t}+h(t)=x(t),\mathrm{with}h(0)=x(0)$, which after expansion with first-order forward Eulerian expansion is $h_t=(1-\alpha_t)h_{t-1}+\alpha _tx_t,\mathrm{with}h_0=x_0$, and $\alpha_t$ is the filter coefficient. And the core formula of our work is $\frac{\mathrm{d}x(t)}{\mathrm{d}t}=-f\left(x(t),I(t),t,\theta\right)x(t)+f\left(x(t),I(t),t,\theta\right)g(x(t),I(t),t ,\theta)$, where **$f\left(x(t),I(t),t,\theta\right)$ is used to adaptively control the degree of filtering and $g(x(t),I(t),t,\theta)$ is used to adaptively control the Lipschitz constant. A semi-implicit Eulerian discretization is used and then solved in continuous time 1s.** Our work is based on Neural ODE unfolding, which is fundamentally different as can be seen from the mathematical formulation and the way it is solved.
>
> 6. In terms of experimental setup, the work you mentioned trains **in a noisy environment by artificially adding Gaussian noise**, which is very different from the basic RL training environment. This training environment setting may result in an algorithm that has a significant advantage when dealing with Gaussian noise and may have a degraded performance when dealing with other noises.
>
> 7. Their work requires preserving the hidden state of the 4-step history for filtering, which makes it **difficult to seamlessly integrate their algorithm into other RL algorithms and frameworks**.
>
> 8. Finally, our work has been submitted once before **2024.08.21 (the date of publication of the work you mentioned)** (the core METHOD is identical), and you can contact us after the review if you need further proof.
>
> In summary, there are no academic integrity issues associated with our work.

---

### Official Review · Reviewer_kjCQ · 2024-11-04

**Soundness:** 3
**Presentation:** 2
**Contribution:** 4
**Rating:** 8
**Confidence:** 3

**Summary:**

This paper introduces a novel neural ODE-based architecture designed for reinforcement learning tasks, specifically addressing the problem of action fluctuation. The authors use ODEs as low-pass filters on network hidden states, with provided theoretical justifications. Their method controls the network's Lipschitz constant in a state-dependent manner, allowing for large actions when needed and smooth actions otherwise. The work demonstrates improved performance over state-of-the-art methods through comprehensive experimentation and ablation studies.

**Strengths:**

1. The novel approach of integrating smoothing into neural ODEs for resolving action fluctuation is an interesting idea.
2. Having state-dependence is shown to be important for performance, which is important knowledge for the community.
3. The comprehensive ablation studies show that the individual choices made are important for the performance of the method.
4. Testing with multiple noise levels on mujoco tasks is a good way of benchmarking such methods, and the results show clear improvements over existing techniques.
5. The authors provide proofs showing how their method can bound the Lipschitz constant.
6. The authors describe how their ODE integration is performed carefully making choices so that the method is not too expensive for practical use.

**Weaknesses:**

## Major issues

1. Limitations discussion hidden in appendix instead of main paper under future work
3. Neural ODE section explanation is unclear, particularly
4. Notation used is often confusing or partially defined:
    - Equation 12 has the variable j which is never quantified, I believe there is a missing summation
    - some undefined terms such as g(.)^max, this is not clear what the max is over
    - In equation 14 l is defined to take one parameter but then takes 2 in the equation
5. "Bionic modeling" terminology is confusing and includes notions of membrane capacitance with arbitrary values (0.4-0.6) which lack justification, and it is unclear if the model aims for biological plausibility or mathematical abstraction
6. The authors mention that the limitation of Kalman filters is that they only function well with Gaussian noise but their tests only include Gaussian noise.

## Minor Issues
1. Line 096: "Multi-layer perception" should be "Multi-layer perceptron"
2. Line 332: "Regular" should be "Regularization"
4. Line 376: DSAC is mentioned as state-of-the-art, but newer methods outperform DSAC, such as TQC and CrossQ, however this does not detract from the contribution of the paper
3. Line 746: "According to Eq. equation 17" should be "According to Eq. 17"
## Suggestions
1. Move limitations section to main paper
2. Add clarity to mathematical notations, such as defining what g(.)^max is
3. Fix equation 12 as it's missing a summation
4. Use h(t) instead of x(t) for hidden states as previous literature does would be clearer
5. Use non-gaussian noise in some tests

**Questions:**

How does this work relate to ODE-based Recurrent Model-free RL for POMDPs (https://arxiv.org/abs/2309.14078)?

---

> ### Author Response · Authors · 2024-11-18
> **[1/2] Rebuttal by Authors**
>
> We appreciate the reviewer for the careful reading of our paper and detailed discussions.
>
> ### **> Major issue 1**
> Your suggestion is spot on, the discussion about Limitation should really be shown in the main text section. We have revised the paper to include that section at the end of page 10.
> ### **> Major issue 2**
> #### **2.1**
> Thanks to your suggestion, we have rewritten Equation 12 as
>
> $\frac{\mathrm{d}x _ {i}}{\mathrm{d}t}=\sum _ {j}\left[-\frac{w _ {ij}}{C _ {\mathrm{m} _ {i}}}\sigma _ {i}\left(x _ {j}\right)x _ {i}+\frac{w _ {ij}}{C _ {\mathrm{m} _ {i}}}\sigma _ {i}\left(x _ {j}\right)\cdot\tanh(h\left(x _ {j},\theta\right))\right]+x _ {\mathrm{leak} _ {i}}$
>
> #### **2.2**
> You can look at line 242. We define $g(x(t), I(t), t, \theta)=\text{tanh}(h(x(t), I(t), t, \theta))$, where tanh stands for a hyperbolic tangent function, and $h(x(t), I(t), t, \theta)$ stands for a neural network. Thus, $g(x(t), I(t), t, \theta)$ takes values in the range (-1, 1).
>
> To describe it further, $g(x(t), I(t), t, \theta)_i$ outputs a scalar $G$ after getting the state input, then $g(x(t), I(t), t, \theta)^{max}_i=|G|, g(x(t), I(t), t, \theta)^{min}_i=-|G|$.
>
> #### **2.3**
> You're right that this part could be misleading. We redefine $\frac{\mathrm{d}x}{\mathrm{d}t} = L(x)$, where $L(x)=l(x(t),x(t+\Delta t))$. The $l$ in the original equation, although it takes only one parameter in the differential equation, is extended after numerical discretization to accept values at two points in time, which is done to achieve an implicit solution.
>
>
> ### **> Major issue 3**
> As you can sense, “Bionic modeling” can be confusing at first glance, but the term actually comes from the work on Neural circuit policies (NCP) enabling auditable autonomy [1]. The researchers in this work **reference the structure of the nervous system of *Caenorhabditis elegans*,** a nematode renowned for its near-optimal neural network architecture and coordinated neural information processing mechanisms. NCP mentions that bionic modeling of neurons using imitation *Caenorhabditis elegans* can enhance their computational capabilities. Therefore, biological concepts such as membrane capacitance are mentioned in our paper as learnable parameters with an initial range of values consistent with those in the NCP.
>
> ### **> Reference**
> [1] Lechner et al. "Neural circuit policies enabling auditable autonomy", Nature machine intelligence, 2020.

---

> ### Author Response · Authors · 2024-11-18
> **[2/2] Rebuttal by Authors**
>
> ### **> Major issue 4**
> Your suggestion is valuable; it is essential to test under non-Gaussian noise, as this aligns with the motivation discussed in introduction. Therefore, we decided to experiment with **uniform noise** as a representative of non-Gaussian noise.
>
> For the MuJoCo environment, we selected four tasks Humanoid-v3, Walker2d-v3, Pusher-v2, and Hopper-v3 for our experiments. Since the state values of different Mujoco tasks vary greatly, we set two levels of uniform noise for the four tasks, as shown in Table1. For the whole tasks, noise is added to all states.
>
> **Table 1 Different levels of uniform noise for different Mujoco tasks.** The numbers in the table represent the range of values for uniform noise.
> | Noise level | Humanoid-v3   | Walker2d-v3   | Pusher-v2     | Hopper-v3     |
> | ----------- | ------------- | ------------- | ------------- | ------- |
> | level 1     | [-0.05, 0.05] | [-0.15, 0.15] | [-0.10, 0.10] | [-0.05, 0.05] |
> | level 2     | [-0.10, 0.10] | [-0.20, 0.20] | [-0.20, 0.20] | [-0.07, 0.07] |
>
> Three different policy networks were used in the experiments, the first one is MLP network, the second one is MLP+LowFilter and the third one is SmODE. **LowFilter is a low-pass filter that utilizes a three-step history state.** The extended Kalman filter is not employed here because it requires state transition and observation equations, as well as noise statistics, which are typically unavailable in general reinforcement learning environments such as MuJoCo. The results, which are the averages of five seeds, are shown in Table 2.
>
> **Table 2 Experimental results.** Average control performance of MLP, MLP+LowFilter, SmODE for different uniform noise levels, where level 1 is on the left column and level 2 is on the right column. The average action fluctuation rate is indicated in parentheses. Results are expressed as mean ± standard deviation of five independent environmental seeds.
> | Network structure | Humanoid-v3                     | Walker2d-v3                     | Pusher-v2                  | Hopper-v3                        |
> | ----------------- | ------------------------------- | ------------------------------- | -------------------------- | -------------------------------- |
> | MLP               | 10004±77 (1.53)  9968±45 (1.77) | 4959±620 (1.70) 964±705 (2.00)  | -76±9 (1.36)  -96±6 (1.57) | 1669±430 (0.73)  1425±346 (0.89) |
> | MLP+LowFilter     | 9428±79 (1.29)  9559±106 (1.41) | 4444±679 (1.27) 4066±877 (1.48) | -44±2 (0.87)  -81±6 (1.26) | 1338±542 (0.51)  1185±582 (0.62) |
> | SmODE             | **10552±36 (0.42) 10478±18 (0.48)** | **5678±607 (0.91) 5585±705 (1.12)** | **-37±4 (0.39)  -42±3 (0.89)** | **3331±492 (0.72)  3249±611 (0.82)** |
>
> Under different levels of uniform noise, SmODE, functioning as a policy network, achieved the **lowest average action fluctuations and the best performance** compared to MLP and MLP+LowFilter in the whole Mujoco tasks. Comparing MLP with MLP+LowFilter, we observe that while incorporating a low-pass filter reduces action fluctuations ratio, it may also lead to performance degradation in the Humanoid-v3, Walker2d-v3, and Hopper-v3 tasks.
>
> ### **> Minor issues**
> Thank you very much for reading the paper so carefully, we have revised all the relevant expressions in the paper, and we believe that these changes can play a very important role in improving the readability and quality of the paper.
>
>
> ### **> Suggestion**
> Your suggestion to use $h(t)$ as hidden states is quite good to be consistent with previous work. However, both $h(t)$ and $x(t)$ have been defined in the current paper, and redefining $h(t)$ as hidden states requires more formulae to be modified, which **may cause trouble to other reviewers.** I hope you can understand that we will not make changes for the time being, and we will **do it uniformly after the review is finished, thank you very much.**
> ### **> Question**
> We reviewed the work you mentioned [2], which aims to address **the challenge of inferring unseen information from raw observations in a partially observable (PO) environment.** Their approach proposes a recursive model based on Neural ODEs, integrated with a model-free reinforcement learning framework. Unseen information is inferred by encoding historical observations, actions, and rewards, and its effectiveness is demonstrated across various PO continuous control and meta-reinforcement learning tasks. The core of the model is the GRU-ODE, which combines gated recurrent units with neural ODEs to manage hidden state transitions and potential state transitions.
>
> While both apply Neural ODE to deep reinforcement learning, this work you mention is quite different from the neural network control non-smoothing problem we are trying to solve. Therefore, the methods devised are  quite different, **but we both agree that Neural ODE is highly robust.**
> ### **> Reference**
> [2] Zhao et al. "ODE-based Recurrent Model-free Reinforcement Learning for POMDPs", NeurIPS, 2023.

---

> ### Comment · Reviewer_kjCQ · 2024-11-19
> **Response to rebuttal**
>
> Thank you for comprehensively addressing most of my concerns. I am confident that the final paper is strong and will update my rating from a 6 to an 8. One more note to the authors, the action fluctuation ratio is not a great metric for measuring the amount noise in the output as it is sensitive to the dimensionality of the action space and weighs many small action changes the same as one large action change. I realize that it's an easy metric to compute and that the work you compare against uses it, but I do believe a Fourier transform based metric better captures the notion of action noise reduction. This comment does not detract from existing strength of the work and I do not expect the authors to redo their analysis.

---

> > ### Author Response · Authors · 2024-11-19
> > **Re: Reviewer kjCQ**
> >
> > Your recognition and suggestions are one of the great motivations for us to continuously improve our work and supplement our experiments. Reviewer DVZD also mentioned this issue, so we have added the relevant experiments. We found another test metric named mean weighted frequency (MWF) in [1] [2].
> >
> > When analyzed in the frequency domain, the unsmoothness of the control policy can be explained by the high-frequency components in the action sequence. Given an action sequence $\{a_{0},a_{1},\cdots a_{T}\}\sim\rho_{\pi}$, the MWF is defined
> >
> > $\zeta(\pi)=\mathbb{E} _ {\{a_0,a_1,\cdots a_T\}\sim\rho_\pi}\left[\sum _ {i=1}^n\frac{\sum _ {j=1}^kA_{ij}f_{ij}}{n\sum _ {j=1}^kA_{ij}}\right],$
> >
> > where $A_{ij}$ and $f_{ij}$ is the $j$-th amplitude and frequency component in the frequency spectrum of the $i$-th dimension action sequence.
> >
> >
> > We selected four robot control tasks, Humanoid-v3, Walker2d-v3, Pusher-v2 and Hopper-v3, and used the MWF indicator for experimental testing. Since the state values of different Mujoco tasks vary greatly, we set two levels of uniform noise for the four tasks, as shown in Table1. For the whole tasks, noise is added to all states.
> >
> > **Table 1 Different levels of uniform noise for different Mujoco tasks.** The numbers in the table represent the range of values for uniform noise.
> > | Noise level | Humanoid-v3   | Walker2d-v3   | Pusher-v2     | Hopper-v3     |
> > | ----------- | ------------- | ------------- | ------------- | ------------- |
> > | level 1     | [-0.05, 0.05] | [-0.15, 0.15] | [-0.10, 0.10] | [-0.05, 0.05] |
> > | level 2     | [-0.10, 0.10] | [-0.20, 0.20] | [-0.20, 0.20] | [-0.07, 0.07] |
> >
> > The comparison network structures used in the experiment are MLP, MLP-LowFilter and SmODE. The results, which are the averages of five seeds, are shown in Table 2.
> >
> > **Table 2 Experimental results.** Average control performance of MLP, MLP-LowFilter, and SmODE for different uniform noise levels, where level 1 is on the left column and level 2 is on the right column. The average mean weighted frequency is indicated in parentheses. Results are expressed as mean ± standard deviation of five independent environmental seeds.
> > | Network structure | Humanoid-v3                             | Walker2d-v3                               | Pusher-v2                          | Hopper-v3                                  |
> > | ----------------- | --------------------------------------- | ----------------------------------------- | ---------------------------------- | ------------------------------------------ |
> > | MLP               | 10004±77 (10.66Hz)  9968±45 (11.69Hz)   | 4959±620 (26.12Hz) 964±705 (28.13Hz)      | -76±9 (4.82Hz)  -96±6 (5.14Hz)     | 1669±430 (24.35Hz)  1425±346 (26.17Hz)     |
> > | MLP+LowFilter     | 9428±79 (9.32Hz)  9559±106 (10.12Hz)    | 4444±679 (22.76Hz) 4066±877 (24.06Hz)     | -44±2 (4.06Hz)  -81±6 (4.58Hz)     | 1338±542 (22.39Hz)  1185±582 (22.67Hz)     |
> > | SmODE             | **10552±36 (7.45Hz) 10478±18 (8.22Hz)** | **5678±607 (20.05Hz) 5585±705 (22.37Hz)** | **-37±4 (3.31Hz)  -42±3 (4.03Hz)** | **3331±492 (23.32Hz)  3249±611 (23.71Hz)** |
> >
> > Finally, we sincerely thank you for taking the time out of your busy schedule to review our paper.
> >
> > ### **> Reference**
> > [1] Mysore et al. "Regularizing Action Policies for Smooth Control  with Reinforcement Learning", IEEE ICRA, 2021.
> >
> > [2] Wang et al. "Smooth Filtering Neural Network for  Reinforcement Learning", IEEE TIV, 2024.

---

### Official Review · Reviewer_6QXf · 2024-11-04

**Soundness:** 3
**Presentation:** 4
**Contribution:** 3
**Rating:** 8
**Confidence:** 3

**Summary:**

To adress the issue of smoothness in a policy in RL the authors propose an ODE-based approach to perform a low-pass filtering of the methods.

**Strengths:**

- The paper deals in an important area of research: finding stable control policies (where smoothness is one aspect) is a relevant area of research
- the derivations are sound and the concepts are explained in a clear way
- overall the paper is written quite well
- the authors perform experiments on different domains
- relevant work is mentioned as far as I can tell

**Weaknesses:**

- One key problem is that the authors motivate their method e.g. by: " Filtering methods like Kalman and extended Kalman filtering Chen et al. (2023) effectively suppress noise and reduce output oscillation by estimating the current state from multi-step historical data. These methods work well with Gaussian noise but struggle with non-Gaussian noise."

However, in the experiment the authors only test on settings with Gaussian noise:
  - MuJoco:  Table 2,3 and 4 specify a Gaussian noise level
  - vehicle trajectory tracking environment: it is unclear what the noise shape is here (perhaps its partial-observable and thus exhibits non-Gaussian noise). In that case the authors should perform an analysis of the shape of stochasticity in this benchmark.

Either way: I would advise either testing again a Filtering method, such as an extended Kalman filter, re-designing the experiments under non-Gaussian settings and/or clearly present how the vehicle trajectory tracking is a RL problem with non-standard noise.

**Questions:**

.

---

> ### Author Response · Authors · 2024-11-16
> **Rebuttal by Authors**
>
> We thank you for the careful reading of our paper and constructive comments in detail.
>
> ### **> Weakness**
> Your suggestion is valuable; it is essential to test under non-Gaussian noise, as this aligns with the motivation discussed in introduction. Therefore, we decided to experiment with **uniform noise** as a representative of non-Gaussian noise.
>
> For the MuJoCo environment, we selected four tasks Humanoid-v3, Walker2d-v3, Pusher-v2, and Hopper-v3 for our experiments. Since the state values of different Mujoco tasks vary greatly, we set two levels of uniform noise for the four tasks, as shown in Table1. For the whole tasks, noise is added to all states.
>
> **Table 1 Different levels of uniform noise for different Mujoco tasks.** The numbers in the table represent the range of values for uniform noise.
> | Noise level | Humanoid-v3   | Walker2d-v3   | Pusher-v2     | Hopper-v3     |
> | ----------- | ------------- | ------------- | ------------- | ------------- |
> | level 1     | [-0.05, 0.05] | [-0.15, 0.15] | [-0.10, 0.10] | [-0.05, 0.05] |
> | level 2     | [-0.10, 0.10] | [-0.20, 0.20] | [-0.20, 0.20] | [-0.07, 0.07] |
>
> Three different policy networks were used in the experiments, the first one is MLP network, the second one is MLP+LowFilter and the third one is SmODE. **LowFilter is a low-pass filter that utilizes a three-step history state.** The extended Kalman filter is not employed here because it requires state transition and observation equations, as well as noise statistics, which are typically unavailable in general reinforcement learning environments such as MuJoCo. The results, which are the averages of five seeds, are shown in Table 2.
>
> **Table 2 Experimental results.** Average control performance of MLP, MLP+LowFilter, SmODE for different uniform noise levels, where level 1 is on the left column and level 2 is on the right column. The average action fluctuation rate is indicated in parentheses. Results are expressed as mean ± standard deviation of five independent environmental seeds.
> | Network structure | Humanoid-v3                     | Walker2d-v3                     | Pusher-v2                  | Hopper-v3                        |
> | ----------------- | ------------------------------- | ------------------------------- | -------------------------- | -------------------------------- |
> | MLP               | 10004±77 (1.53)  9968±45 (1.77) | 4959±620 (1.70) 964±705 (2.00)  | -76±9 (1.36)  -96±6 (1.57) | 1669±430 (0.73)  1425±346 (0.89) |
> | MLP+LowFilter     | 9428±79 (1.29)  9559±106 (1.41) | 4444±679 (1.27) 4066±877 (1.48) | -44±2 (0.87)  -81±6 (1.26) | 1338±542 (0.51)  1185±582 (0.62) |
> | SmODE             | **10552±36 (0.42) 10478±18 (0.48)** | **5678±607 (0.91) 5585±705 (1.12)** | **-37±4 (0.39)  -42±3 (0.89)** | **3331±492 (0.72)  3249±611 (0.82)** |
>
> Under different levels of uniform noise, SmODE, functioning as a policy network, achieved the **lowest average action fluctuations and the best performance** compared to MLP and MLP+LowFilter in the whole Mujoco tasks. Comparing MLP with MLP+LowFilter, we observe that while incorporating a low-pass filter reduces action fluctuations ratio, it may also lead to performance degradation in the Humanoid-v3, Walker2d-v3, and Hopper-v3 tasks.
>
> I apologize if my previous explanation was unclear. The vehicle tracking problem is not a POMDP problem, as we have access to the desired trajectory points over a longer time horizon as part of the observation, along with the vehicle's state information.
>
> ---
>
> There is a brief discussion of using filters in page 6 of CAPS [1]. The discussion emphasizes that neural network-based control behaves fundamentally differently from traditional controllers. Neural network policies are typically not trained with integrated filters, **so deploying them with a filter alters the dynamic response expected by the network, potentially leading to anomalous behavior.** The paper further suggests that naively introducing filters into the training pipeline, without also incorporating the corresponding state history, can result in catastrophic learning failures. This likely occurs because **it disrupts the Markov assumption, which is central to reinforcement learning theory.** On the other hand, compensating for this by including the state history to maintain the Markov assumption could significantly increase the representation complexity of the problem, due to the higher-dimensional input states. Our SmODE also achieves very good performance and smoothing results even under conditions that do not require direct input of historical states, suggesting that **such compromises are unnecessary.**
>
> ### **> Reference**
> [1] Mysore et al. "Regularizing Action Policies for Smooth Control  with Reinforcement Learning," IEEE ICRA, 2021.

---

> ### Comment · Reviewer_6QXf · 2024-11-21
>
> thank you for the additional experiments. I updated my score.
>
> However: When you argue that your method works better than related work under non-Gaussian noise -- using uniform noise is a  bit obvious. Thats still "nice" noise, i.e. unimodal, "Gaussian-like", compared to  non-symmetric (think exponential) or multi-modal distributions.
>
> To really bring the point home an alternative way is to make the benchmark partially-observable (e.g masking one or a small subset of sensors). By that you can achieve complex (and state-dependent) noise.

---

> > ### Author Response · Authors · 2024-11-21
> > **Re: Reviewer 6QXf**
> >
> > Thank you very much for your suggestions on partially sizable conditions or the use of more complex noise. We will test it later in our work using the method you suggested. Thank you again for taking time out of your busy schedule to review our paper.

---

### Meta-Review · Area_Chair_ytTo · 2024-12-18

**Metareview:**

This work proposes a bio-inspired NeuralODE with a bounded Lipschitz constant, which can serve as a neuron in a neural network for approximating smooth functions. The authors leverage this SmODE network as a policy network in an actor-critic algorithm for RL, enabling smooth action outputs in a noisy environment. The paper presents relevant theoretical results alongside numerical experiments that demonstrate the utility of the proposed approach.

Three out of four reviewers rated this work highly, commending the well-explained motivation and extensive empirical validation. However, concerns remain regarding notational clarity and theoretical rigor, particularly the insufficient description of the core neuron model underlying the SmODE network, even after revisions. Additionally, several issues were identified during the meta-review:

- The dimensions of x, and the coefficient functions f, g, and I in the ODE are not clearly specified. Clarifying whether these are scalars, vectors, or matrices is crucial, as some arguments in the theoretical proofs appear problematic without this information. Given the importance of demonstrating control over the ODE neuron’s Lipschitz constant, the proof should be carefully revised.

- In Fig 1 panel (b), the variable t is used for bothe the discrete time index in RL (as in s(t) and a(t) on both sides) and the continuous time index in the NeuralODE process. This overlap is confusing, and using a different variable for the RL time index would improve clarity.

Given the compelling results and limitations in the rigorous of the presentation, the AC recommends accpetance for poster.  The AC encourages the authors to address the noted shortcomings in a future revision, particularly by improving the clarity of notations and the rigor of theoretical arguments.

**Additional Comments On Reviewer Discussion:**

The reviewers raised several concerns, including the clarification of notations, the choice of the action fluctuation ratio as a metric for smoothness, and the limited setting of Gaussian noise, which could be addressed by Kalman filtering as a baseline. The authors made significant efforts to address these points, providing additional numerical results and revising the manuscript to improve clarity. Most concerns were adequately resolved, leading one reviewer to update the score.

One reviewer raised concerns about significant overlap with a recent publication in *IEEE TIV*. The authors clarified that, while both works share a high-level goal, the proposed SmoothODE network significantly reduces network complexity and employs more realistic training settings. These distinctions suggest that the contribution can be considered independent and novel.

---

### Decision · Program_Chairs · 2025-01-22

Accept (Spotlight)